# OmniPainter: Global-Local Temporally Consistent Video Inpainting Diffusion Model

## Abstract

Video inpainting methods often fail to resolve the inherent trade-off between long-term global consistency and short-term local smoothness, leading to artifacts such as contextual drift and flickering. We introduce OmniPainter, a latent diffusion framework designed to address this limitation. Our framework is built on two core innovations: a Flow-Guided Ternary Control mechanism for superior structural fidelity, and a novel Adaptive Global-Local Guidance strategy. This guidance strategy dynamically blends two complementary guidance scores at each denoising step: an autoregressive score to enforce local transitional smoothness, and a hierarchical score to maintain long-range global coherence. The blending weight is determined by a function of both the video's motion dynamics and the current diffusion timestep. This adaptive blending allows the model to prioritize global structure during the early stages of generation and then shift focus to local continuity during later refinement stages, thereby achieving a robust temporal equilibrium. Extensive experiments confirm that OmniPainter significantly outperforms state-of-the-art methods, setting a new standard for temporally consistent video restoration.

## 1 Introduction

Recent advances in deep learning have revolutionized video inpainting, enabling unprecedented restoration of missing or occluded regions in video content Lee et al. (2019); Gao et al. (2020); Zeng et al. (2020); Zou et al. (2021); Liu et al. (2021). While early techniques reliant on optical flow and patch-based propagation Chang et al. (2019); Li et al. (2022b) often faltered with complex motion, modern approaches leveraging generative models like Diffusion Models Ho et al. (2020); Li et al. (2025); Zhang et al. (2024b); Wan et al. (2024) have achieved significant breakthroughs. Nevertheless, a critical challenge remains unresolved: ensuring robust temporal consistency across extended video sequences.

This challenge involves a difficult trade-off between two distinct requirements, which we term *global-local temporal consistency*. *Global consistency* refers to maintaining the identity, appearance, and structure of objects and scenes over long durations. In contrast, *local consistency* pertains to generating smooth, flicker-free transitions between adjacent frames. Many state-of-the-art methods, including those using Transformers Guo et al. (2023); Blattmann et al. (2023), excel at one aspect but often at the expense of the other. For instance, autoregressive models Cui et al. (2024) can produce smooth local transitions but are prone to error accumulation, leading to a gradual drift in global context. Conversely, hierarchical or keyframe-based approaches Li et al. (2025) may preserve global structure but can introduce subtle discontinuities between interpolated segments (See Figure 1). Consequently, no existing approach has effectively reconciled these conflicting demands.

To address this critical gap, we propose *OmniPainter*, a novel video inpainting framework based on a Latent Diffusion Model (LDM). OmniPainter is specifically designed to achieve superior global-local temporal consistency by introducing two synergistic components that manage spatial details and temporal dynamics in a more sophisticated manner. Our main contributions are:

**Flow-Guided Ternary Control:** We introduce a novel ternary control mask that provides nuanced spatial guidance to the diffusion model. Instead of treating all missing pixels equally, we leverage a reliable optical flow prior Teed & Deng (2020) to partition masked regions into three categories: those requiring full *inpainting* from scratch, those to be *preserved*, and crucially, those to be *refined*.

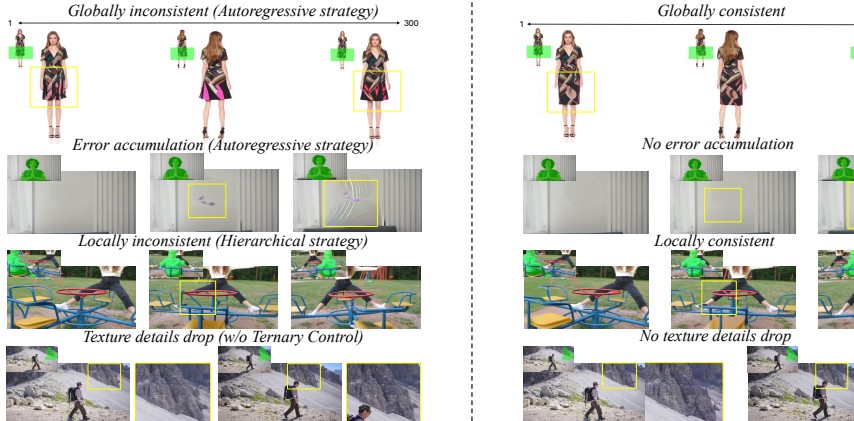

Figure 1: Existing strategies often suffer from inherent limitations in maintaining temporal consistency and generating reliable results (left). In contrast, our proposed framework, *OmniPainter*, is designed to overcome these shortcomings by achieving temporally consistent and robust video inpainting.

This refinement capability allows the model to utilize valuable high-frequency textures from the flow-warped result while correcting its inaccuracies, significantly enhancing structural stability and detail preservation.

**Adaptive Global-Local Guidance:** To explicitly address the temporal consistency dilemma, we propose a novel guidance mechanism that dynamically blends two complementary strategies during the denoising process. *Autoregressive (AR) Guidance* enforces smooth continuity between adjacent groups of frames, excelling in high-motion scenarios. *Hierarchical (HR) Guidance* ensures long-range coherence by conditioning on distant keyframes, proving effective for maintaining global context. The blending of these two scores is adaptively controlled by both the video's motion magnitude and the current diffusion timestep. This allows OmniPainter to prioritize global structure (via HR) in the early stages of generation and then shift focus to local smoothness (via AR) in the final refinement stages, achieving a robust temporal equilibrium.

Extensive experiments demonstrate that the combination of these innovations enables OmniPainter to significantly outperform existing methods in both visual quality and temporal coherence. Our framework substantially mitigates flickering and long-term drift, paving the way for more reliable and visually consistent video restoration in challenging real-world scenarios.

## 2 RELATED WORK

**Video Inpainting.** Video inpainting has advanced considerably, yet existing approaches still face critical limitations. Early methods relying on patch-based propagation Chang et al. (2019) often failed when faced with complex motion. To address this, optical flow-guided models such as E2FGVI Li et al. (2022b), FGT++ Zhang et al. (2024a), and ProPainter Zhou et al. (2023) were introduced to enhance motion coherence by explicitly warping content from reference frames. While effective, their performance is highly dependent on the accuracy of the flow estimation. They often struggle in low-motion scenes where flow is unreliable and, crucially, treat the flow-warped prior as a fixed guide, failing to correct its inherent inaccuracies. Our work addresses this by introducing a *Ternary Control* mechanism that allows the model to selectively refine rather than simply follow the flow-based prior. On the other hand, Transformer-based architectures like STTN Zeng et al. (2020) and FuseFormer Liu et al. (2021) leverage attention to capture temporal dependencies. However, their fixed receptive fields often prevent them from enforcing consistent object appearance and structure over long sequences. More recent diffusion-based methods Li et al. (2025); Gu et al. (2024); Zhang et al. (2024b); Wan et al. (2024); Zi et al. (2024); Lee et al. (2024); Wang et al. (2023b) have demonstrated powerful generative capabilities but still fall short in resolving the fundamental tension between maintaining long-term global context and short-term local smoothness.

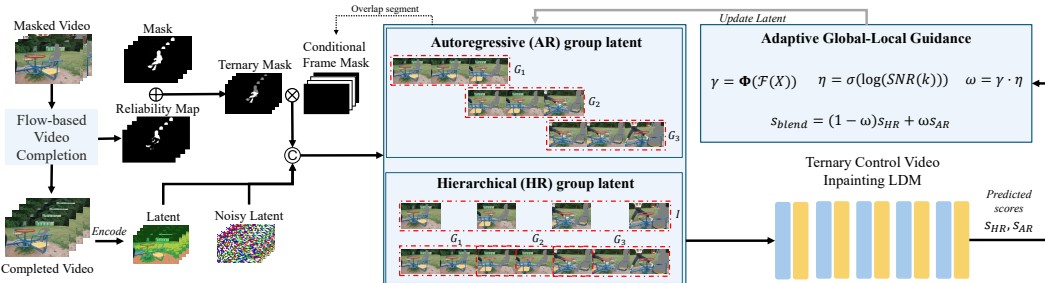

Figure 2: Overview of our OmniPainter framework. Here, ⓒ denotes concatenation, ⊕ represents masked blending, and ⊗ stands for pixel-wise product.

**Long Video Generation.** The challenge of maintaining temporal consistency is also central to long video generation. Methodologies in this field offer valuable insights into managing temporal dynamics. Autoregressive models Xie et al. (2024); Li et al. (2024); Cui et al. (2024); Valevski et al. (2024); Chung et al. (2024); Yang et al. (2023) generate frames sequentially, which excels at creating smooth local transitions. However, this approach is susceptible to error accumulation, causing the global context to drift over time. In our framework, we harness the principle of autoregression for our *AR Guidance* to ensure local continuity, but mitigate its drift problem by integrating it into a broader guidance system. Conversely, hierarchical strategies Zhou et al. (2025); Wang et al. (2023a); Yin et al. (2023); Bar-Tal et al. (2024); Harvey et al. (2022) attempt to secure global consistency by generating keyframes first and then filling in the gaps. While this helps preserve long-range dependencies, it can result in abrupt or unnatural transitions between segments.

Unlike these prior works, which typically commit to one strategy, OmniPainter resolves this dilemma through its **Adaptive Global-Local Guidance**. Our framework synthesizes the strengths of both autoregressive and hierarchical approaches, dynamically blending them based on scene motion and the diffusion process itself. This allows us to achieve both global and local temporal consistency, a combination that has remained elusive in the specific context of video inpainting.

## 3 METHOD

Our video inpainting framework, shown in Figure 2, consists of two core modules. First, the **Ternary Control Video Inpainting LDM** leverages optical-flow warping and a ternary mask to pre-fill reliably moving regions, preserve fine texture, and enforce temporal coherence and local texture details. Second, the **Global–Local Guidance** integrated with conditional training dynamically blends autoregressive and hierarchical score estimates based on motion strength and diffusion timestep; by alternately conditioning on previous groups and sparse keyframes during training, the model learns both smooth group-level transitions and sparse-to-dense interpolations, enabling efficient high-quality inpainting of long video sequences.

### 3.1 TERNARY CONTROL VIDEO INPAINTING LDM

We consider a masked video sequence $X = [x_1, \ldots, x_n]$ with corresponding binary masks $M = [m_1, \ldots, m_n]$, where $x_t \in \mathbb{R}^{H \times W \times 3}$ is the $t$-th frame and $m_t \in \{0, 1\}^{H \times W \times 1}$ is its mask. To enforce temporal coherence and mitigate artifacts, we first leverage reliable motion information from adjacent frames. Following the image propagation strategy of PropPainter Zhou et al. (2023), we compute a pre-filled frame $\hat{x}_t$ using the optical flow $f_{t-1 \to t}$ and a reliability map $A_r \in \{0, 1\}^{H \times W}$:

$$\hat{x}_t = \mathcal{W}(x_{t-1}, f_{t-1 \to t}) \odot A_r + x_t \odot (1 - A_r), \tag{1}$$

where $\mathcal{W}(\cdot, \cdot)$ is the backward warping operator and $\odot$ denotes element-wise multiplication. The reliability map $A_r$ identifies regions where the flow estimation is confident. This step pre-fills temporally consistent regions, reducing the generative load on the diffusion model for predictable areas.

However, optical flow can be imperfect, introducing artifacts or inaccuracies (See Figure 6). Simultaneously, these warped results contain invaluable high-frequency texture details that, if discarded,

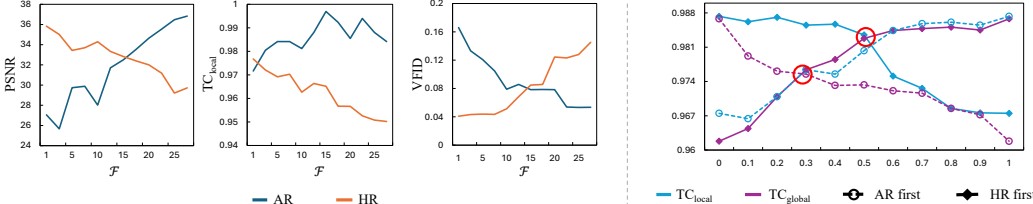

Figure 3: Left: PSNR, $\text{TC}_{\text{local}}$, and VFID as functions of optical flow magnitude $\mathcal{F}$ (x-axis) under AR-only and HR-only guidance. The two guidance modes exhibit contrasting performance trends with increasing flow magnitude. Right: Evolution of $\text{TC}_{\text{local}}$ and $\text{TC}_{\text{global}}$ (shared y-axis). The x-axis denotes the fraction of total denoising steps assigned to the leading guidance modality. AR-first and HR-first represent schedules starting with AR or HR, respectively, as their proportional coverage increases. The intersection point where performance trends cross is marked with a red circle; notably, at this point, the HR-first strategy exhibits relatively higher performance across both metrics. The superior consistency of the HR-first strategy motivates our adaptive temporal blending approach.

force the diffusion model to hallucinate textures from scratch, often leading to blurriness (See Figure 1). To balance these aspects, we introduce a **ternary control mechanism** that partitions the masked regions into two distinct types: regions to be fully *inpainted* and regions to be *refined*.

This is implemented via a ternary control mask $\hat{m}_t$, which designates one of three roles for each pixel $j$:

$$\hat{m}_t(j) = \begin{cases} 1.0 & \text{if } m_t(j) = 1 \text{ and } A_r(j) = 0 \\ \beta & \text{if } m_t(j) = 1 \text{ and } A_r(j) = 1 \\ 0.0 & \text{if } m_t(j) = 0 \end{cases} \tag{2}$$

Here, $\beta \in [0, 1]$ is a hyperparameter that controls the influence of the warped prior. A value of 1 indicates a pure inpainting task, whereas a smaller $\beta$ instructs the model to treat the warped result $\mathcal{W}(x_{t-1}, \cdot)$ as a strong but imperfect prior to be refined. This ternary mask provides a more nuanced spatial guidance than a simple binary mask.

Frames and the ternary control masks are encoded into the latent space of a pretrained VAE $\mathcal{E}(\cdot)$. Let $z_t = \mathcal{E}(x_t)$ and $\hat{m}_t^{\text{lat}} = \text{Resize}(\hat{m}_t)$. The inpainting task is then formulated as learning the conditional score of the latent distribution:

$$s_\theta(\mathbf{y}^{(k)} \mid \mathbf{z}, \hat{\mathbf{m}}) \approx \nabla_{\mathbf{y}^{(k)}} \log p_k(\mathbf{y}^{(k)} \mid \mathbf{z}, \hat{\mathbf{m}}), \tag{3}$$

where $\mathbf{y}^{(k)}$ is the noisy latent representation of the ground-truth video at diffusion step $k$, $\mathbf{z} = [\mathcal{E}(\hat{x}_1), \ldots, \mathcal{E}(\hat{x}_n)]$ is the masked latent video, and $\hat{\mathbf{m}} = [\hat{m}_1^{\text{lat}}, \ldots, \hat{m}_n^{\text{lat}}]$ is the latent control mask sequence.

We train the score network using a simplified DDPM Ho et al. (2020) objective, which regresses to the noise $\epsilon$ added during the forward noising process $\mathbf{y}^{(k)} = \sqrt{\bar{\alpha}_k} \mathbf{y}^{(0)} + \sqrt{1 - \bar{\alpha}_k} \epsilon$, where $\epsilon \sim \mathcal{N}(0, I)$. The objective is:

$$\mathcal{L}_\theta = \mathbb{E}_{\mathbf{y}^{(0)}, \epsilon, k} \left[ \left\| \epsilon - \epsilon_\theta(\mathbf{y}^{(k)}, k, \mathbf{z}, \hat{\mathbf{m}}) \right\|^2 \right], \tag{4}$$

where $\epsilon_\theta$ is the noise prediction network. In inference, the final inpainted frame $\tilde{x}_t$ is obtained by iteratively denoising with the learned score and decoding the result $\tilde{z}_t$ back into the pixel space via $\tilde{x}_t = \mathcal{D}(\tilde{z}_t)$. This ternary-control framework anchors the diffusion process to accurate local details from optical flow, yielding temporally coherent inpainting that faithfully preserves both textures and structure. For notational brevity, we omit $\mathbf{z}$ and $\hat{\mathbf{m}}$ in the following equations.

### 3.2 GLOBAL–LOCAL GUIDANCE

In long-video inpainting, generating reliable per-step scores for all frames at once is impractical. We therefore divide the $n$-frame sequence into contiguous subgroups and employ two complementary guidance scores: an *Autoregressive (AR)* score for local continuity and a *Hierarchical (HR)* score for global coherence.

**Autoregressive Score (AR).** AR guidance enforces smooth transitions between adjacent groups, $G_j$ and $G_{j-1}$. For each frame index $t \in G_j$, the generation process is conditioned on an prediction of the previous group, $\hat{\mathbf{y}}_{G_{j-1}}^{(0)}$. The AR guidance score is then defined by conditioning on this efficiently-obtained prediction:

$$s_{\mathrm{AR}}(t, k) = \nabla_{\mathbf{y}_t^{(k)}} \log p_k\big(\mathbf{y}_t^{(k)} \mid \hat{\mathbf{y}}_{G_{j-1}}^{(0)}\big). \tag{5}$$

Operationally, the conditioning set $\hat{\mathbf{y}}_{G_{j-1}}^{(0)}$ is temporally concatenated with the masked latent $\mathbf{z}_t$, and its corresponding control mask region is set to zero.

**Hierarchical Score (HR).** HR guidance guarantees long-range consistency via a sparse strategy. For a frame $t$ within an interval defined by keyframes $[ih, (i+1)h]$, we refine it by conditioning on the predictions of the two nearest coarse latents, $\hat{y}_{ih}^{(0)}$ and $\hat{y}_{(i+1)h}^{(0)}$:

$$s_{\mathrm{HR}}(t, k) = \nabla_{y_t^{(k)}} \log p_k\big(y_t^{(k)} \mid \hat{y}_{ih}^{(0)}, \hat{y}_{(i+1)h}^{(0)}\big). \tag{6}$$

These two conditioning latents are concatenated with the target frame's masked latent to serve as input to the model. This approach ensures distant frames remain coherent, but interpolation can weaken connectivity under rapid motion.

Our guidance framework leverages intermediate clean latent predictions for conditioning during inference, rather than the final, fully denoised outputs. Following Song et al. (2021), these predictions are estimated directly from the noisy latent state in a single step. For both AR and HR scores, the conditioning latents are estimations derived from the current denoising state, $k$. This strategy provides a dual benefit: it not only enables the adaptive fusion of our guidance scores as the conditions themselves evolve with the denoising process, but also gives the model the flexibility to achieve continuous global refinement throughout the denoising trajectory by utilizing relatively loose constraints.

**Adaptive Guidance.** Neither AR nor HR guidance alone suffices across all video conditions. Our preliminary analysis, shown in Figure 3, reveals two key findings that motivate our adaptive approach. First (Left Panel), AR guidance excels in high-motion regimes, while HR guidance is superior in low-motion contexts. Second (Right Panel), a scheduling strategy that applies HR guidance in the early stages of diffusion and AR in the later stages ("HR first") yields substantially better temporal consistency than the reverse. Note: The diffusion process transitions from a noisy (early stages) to a clean (later stages).

Based on these findings, we designed an adaptive guidance mechanism that dynamically blends AR and HR scores. The blending is controlled by two factors: (1) the video's overall motion dynamics, and (2) the Signal-to-Noise Ratio (SNR), which inherently represents the current stage of the diffusion process.

First, we quantify scene dynamics using the average optical-flow magnitude, $\mathcal{F}(X)$, computed with the RAFT Teed & Deng (2020). The final blending weight, $\omega$, is formulated as a product of a motion-dependent weight, $\gamma$, and a denoising-step-dependent weight, $\eta$. For the motion weight $\gamma$, we employ a bounded exponential function, $\Phi(x) = 1 - e^{-\lambda x}$, which maps the non-negative flow magnitude to a value in $[0, 1)$. The hyperparameter $\lambda$ controls the sensitivity to motion magnitude. For the step-dependent weight $\eta$, we use the sigmoid function, $\sigma(\cdot)$, applied to the logarithm of the SNR, defined as $\mathrm{SNR}(k) = \bar{\alpha}_k / (1 - \bar{\alpha}_k)$. The weights are thus explicitly defined as:

$$\gamma = \Phi\big(\mathcal{F}(X)\big), \quad \eta = \sigma\big(\log(\mathrm{SNR}(k))\big), \quad \omega = \gamma \cdot \eta.$$

This formulation directly implements our finding. With the adaptive weight $\omega$ defined, we compute the blended conditional score:

$$s_{\mathrm{blend}}(t, k) = (1 - \omega) s_{\mathrm{HR}}(t, k) + \omega s_{\mathrm{AR}}(t, k). \tag{7}$$

By dynamically blending the two scores, our adaptive guidance mechanism generates inpainting results that are both locally and globally temporally consistent.

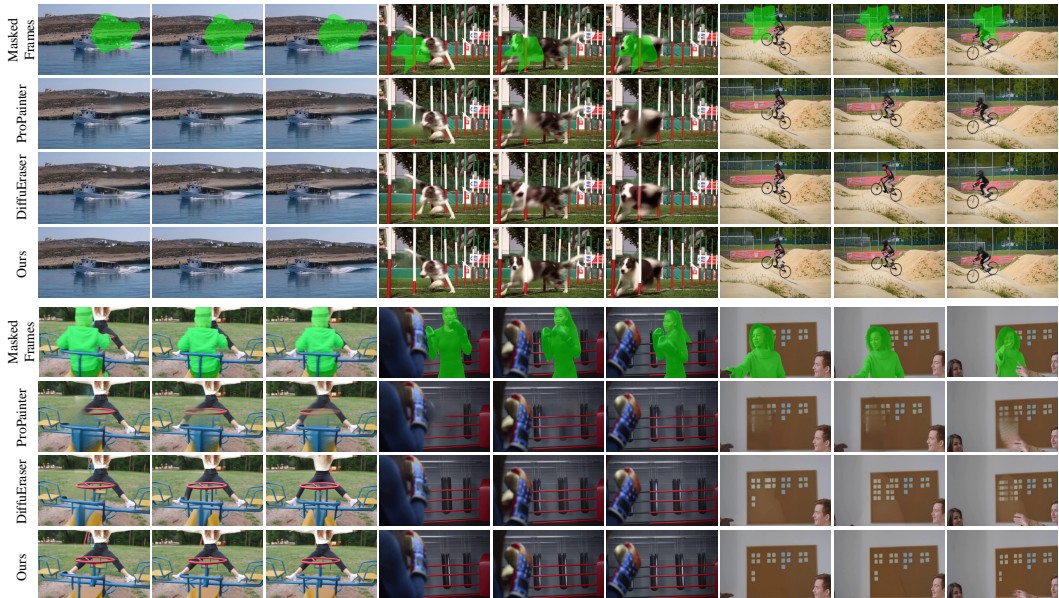

Figure 4: Qualitative comparisons with state-of-the-art methods reveal the strengths of our approach. (Top) In the video completion task, our method preserves intricate textures and ensures smooth transitions between frames, resulting in a highly coherent output. (Bottom) For object removal, our approach seamlessly eliminates the target object while maintaining detailed background structures and stable motion. In both scenarios, our method surpasses existing techniques in visual quality and temporal consistency.

**Global-Local Conditional Training** As detailed in Eq. 4, our noise prediction network $\epsilon_\theta(\mathbf{y}^{(k)}, k, C, \mathbf{z}, \hat{\mathbf{m}})$ is designed to accept a conditioning set $C$. To accommodate both Autoregressive (AR) and Hierarchical (HR) guidance within a single model, $C$ is dynamically sampled at each training iteration:

$$C = \begin{cases} \{\hat{\mathbf{y}}^{(0)}_{G_{j-1}}\}, & \text{with probability } 0.5 \\ \{\hat{\mathbf{y}}^{(0)}_{ih}, \hat{\mathbf{y}}^{(0)}_{(i+1)h}\}, & \text{with probability } 0.5 \end{cases}$$

where $\hat{\mathbf{y}}^{(0)}$ denotes the estimated clean latents for the previous group $G_{j-1}$ or the two nearest sparse keyframes $\{ih, (i+1)h\}$.

The resulting conditional loss is:

$$\mathcal{L}_{\text{cond}} = \mathbb{E}_{\mathbf{y}, \epsilon, k, C} \big\| \epsilon - \epsilon_\theta\big(\mathbf{y}^{(k)}, k, C, \mathbf{z}, \hat{\mathbf{m}}\big) \big\|^2. \tag{8}$$

By randomly alternating between AR and HR conditioning structures during training, a single model learns to perform both local-group continuity and sparse-to-dense interpolation. At inference, this unified model provides the per-frame conditional scores required for our adaptive guidance.

## 4 EXPERIMENTS

**Implementation Details.** Our Latent Diffusion Model (LDM) was trained on a combined dataset from YouTube-VOS Xu et al. (2018) and the self-collected dataset (see details in Appendix A.2), with all content resized to a resolution of $512 \times 512$. We initialize our model using weights from the Stable Diffusion v1.5 inpainting model Rombach et al. (2022) and integrate a motion module from AnimateDiff Guo et al. (2023). During the training phase, the original Stable Diffusion layers were kept frozen. The model was trained for 50,000 iterations with a batch size of 32 with frame interval 16 and using randomly generated masks (details are provided in Appendix A.3). For inference, we employ Latent Consistency Models (LCM) Luo et al. (2023) to accelerate the process to 8 steps.

| Models | YouTube-VOS | | | | | DAVIS | | | | | UBC Fashion | | |
|---|---|---|---|---|---|---|---|---|---|---|---|---|---|
| | PSNR ↑ | SSIM ↑ | VFID ↓ | $E^*_{\text{warp}}$ ↓ | $\text{TC}_{\text{local}}$ ↑ | PSNR ↑ | SSIM ↑ | VFID ↓ | $E^*_{\text{warp}}$ ↓ | $\text{TC}_{\text{local}}$ ↑ | VFID ↓ | $\text{TC}_{\text{local}}$ ↑ | $\text{TC}_{\text{global}}$ ↑ |
| DFVI | 29.16 | 0.9429 | 0.066 | 1.651 | - | 28.81 | 0.9404 | 0.187 | 1.596 | - | - | - | - |
| CPNet | 31.58 | 0.9607 | 0.071 | 1.622 | - | 30.28 | 0.9521 | 0.182 | 1.521 | - | - | - | - |
| FGVC | 29.67 | 0.9403 | 0.064 | 1.163 | - | 30.80 | 0.9497 | 0.165 | 1.571 | - | - | - | - |
| STTN | 32.34 | 0.9655 | 0.053 | 1.061 | - | 30.61 | 0.9560 | 0.149 | 1.438 | - | - | - | - |
| TSAM | 30.22 | 0.9468 | 0.070 | 1.014 | - | 30.67 | 0.9521 | 0.146 | 1.235 | - | - | - | - |
| FuseFormer | 33.32 | 0.9681 | 0.053 | 1.053 | - | 32.59 | 0.9701 | 0.137 | 1.349 | - | - | - | - |
| ISVI | 30.34 | 0.9458 | 0.077 | 1.008 | - | 32.17 | 0.9588 | 0.189 | 1.291 | - | - | - | - |
| FGT | 32.17 | 0.9599 | 0.054 | 1.025 | - | 32.86 | 0.9650 | 0.129 | 1.323 | - | - | - | - |
| E² FGVI | 33.71 | 0.9700 | 0.046 | 1.013 | - | 33.01 | 0.9721 | 0.116 | 1.289 | - | - | - | - |
| ProPainter | 34.43 | 0.9735 | 0.042 | 0.974 | 0.990 | 34.47 | 0.9776 | 0.098 | 1.187 | 0.9760 | 0.248 | 0.9792 | 0.9812 |
| DiffuEraser | 32.52 | 0.9706 | 0.062 | 1.013 | 0.988 | 32.99 | 0.9752 | 0.115 | 1.234 | 0.9756 | 0.055 | 0.9712 | 0.9759 |
| Ours | **35.66** | **0.9878** | **0.040** | **0.955** | **0.993** | **35.84** | **0.9896** | **0.088** | **1.0997** | **0.9766** | **0.042** | **0.9845** | **0.9861** |

Table 1: Quantitative comparisons on YouTube-VOS Xu et al. (2018), DAVIS Perazzi et al. (2016) and UBC Fashion Zablotskaia et al. (2019) datasets. The best is marked in bold. $E^*_{warp}$ denotes $(E_{warp} \times 10^{-3})$. All methods are evaluated following their default settings.

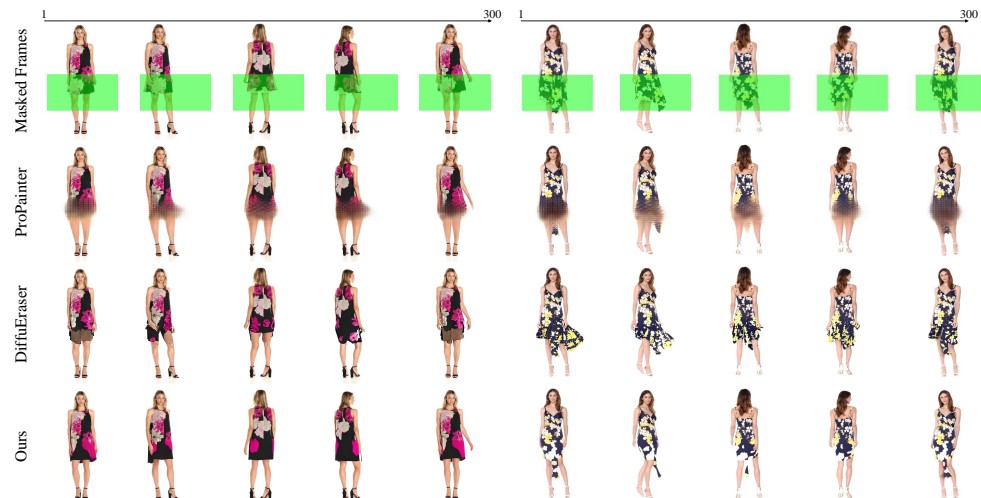

Figure 5: Qualitative comparison with state-of-the-art methods on long video (frame 1 to 300). Our method effectively preserves global-local temporal consistency.

## 4.1 COMPARISONS

**Datasets and Metrics.** For quantitative and qualitative evaluation, we use the DAVIS Perazzi et al. (2016), YouTube-VOS Xu et al. (2018) test set, and UBC Fashion Zablotskaia et al. (2019) datasets. We employ widely adopted metrics to evaluate reconstruction quality and perceptual realism. In our reporting, the arrows ↑ and ↓ indicate that higher and lower values represent better performance, respectively. Specifically, we report PSNR and SSIM Wang et al. (2004) for low-level reconstruction fidelity, and VFID Wang et al. (2018) to measure the perceptual similarity between the generated and ground truth distributions. To assess temporal stability, we utilize the flow warping error ($E_{\text{warp}}$) Lai et al. (2018) for pixel-level consistency. For local semantic continuity, we compute $\text{TC}_{\text{local}}$, defined as the average cosine similarity between CLIP Radford et al. (2021) features of adjacent frames Zhang et al. (2024b). Furthermore, to evaluate long-range semantic stability, particularly for videos with recurring visual patterns (e.g., UBC Fashion), we introduce a global consistency metric ($\text{TC}_{\text{global}}$). This metric measures how well the model maintains appearance consistency across non-adjacent frames representing the same visual content. We first identify a set of frame pairs $\mathcal{S}$ from the ground truth video that maximize CLIP similarity (representing recurring moments). We then calculate $\text{TC}_{\text{global}}$ as the average similarity of these identified pairs in the generated video: $\text{TC}_{\text{global}} = \frac{1}{|\mathcal{S}|} \sum_{(i,j)\in\mathcal{S}} \text{sim}(\text{CLIP}(\hat{V}_i), \text{CLIP}(\hat{V}_j))$, where $\text{sim}(\cdot)$ denotes cosine similarity.

**Quantitative Comparison.** We conduct a comprehensive quantitative comparison of our approach against 11 state-of-the-art methods: DFVI Xu et al. (2019), CPNet Lee et al. (2019), FGVC Gao et al. (2020), STTN Zeng et al. (2020), TSAM Zou et al. (2021), Fuseformer Liu et al. (2021), ISVI Zhang et al. (2022b), FGT Zhang et al. (2022a), E2FGVI Li et al. (2022b), ProPainter Zhou

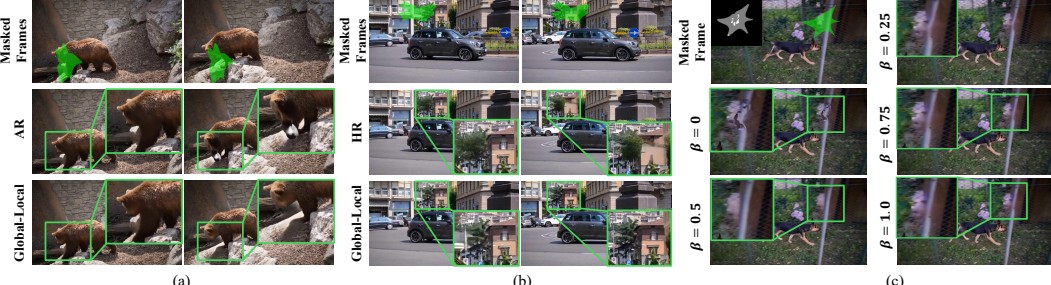

Figure 6: (a) Comparison between using only Autoregressive (AR) guidance and our full Global-Local Guidance. The AR-only approach leads to noticeable texture drift over time. (b) Comparison between using only Hierarchical Score (HR) guidance and our full approach. The HR-only method fails to propagate textures correctly according to motion, resulting in distortion. In contrast, our Global-Local Guidance yields stable, high-quality results in both scenarios. (c) Qualitative comparison demonstrating the effect of the Ternary Mask refinement weight, $\beta$. The corresponding ternary mask is visualized in the top-left corner of each masked frame, showing how our approach effectively corrects structural errors caused by inaccurate flow estimation.

et al. (2023), and DiffuEraser Li et al. (2025). The evaluation is performed on the YouTube-VOS and DAVIS datasets, following the standard settings for each method, while the UBC Fashion dataset is used for our specialized temporal consistency evaluation. In our experiments, we set the key hyperparameters $\beta = 0.5$ and $\lambda = 0.25$. As shown in Table 1, our method decisively outperforms all competing techniques in every evaluation metric. Further validation through a user study is provided in Appendix A.5.

**Qualitative Comparison.** We provide a qualitative comparison against state-of-the-art methods, ProPainter and DiffuEraser, in Figure 4. For standard inpainting (top), ProPainter often generates blurry artifacts in low-motion scenes, while DiffuEraser suffers from inconsistent patterns and temporal flickering. For the object removal task (bottom), where masks are generated using SAM2 Ravi et al. (2024), both competing methods fail to preserve background details, resulting in blur and distortion.

In contrast, our method effectively integrates a flow-based prior with our adaptive guidance strategy. This approach produces stable and coherent results, accurately preserving background patterns and yielding significantly higher visual fidelity in both tasks.

**Long-range Temporal Consistency.** To evaluate the ability of the methods to maintain visual pattern over long video lengths, we conducted an additional qualitative comparison using the UBC Fashion Zablotskaia et al. (2019) dataset. This dataset comprises videos in which a fashion model completes a full rotation from a frontal view back to a frontal view, making it ideal for assessing long-term consistency. As shown in Figure 5, ProPainter fails to preserve any distinct patterns, resulting in a blurred output, while DiffuEraser generates well-defined patterns, but loses consistency through the sequence. In contrast, our method consistently maintains the visual context, preserving the same visual pattern in entire sequence.

## 4.2 ABLATION STUDY

We conduct a series of ablation studies on the DAVIS dataset to rigorously evaluate the core components of our proposed framework: Flow-Guided Ternary Control and the Adaptive Global-Local Guidance mechanism.

**Analysis of the Ternary Mask.** We evaluate the effectiveness of our Flow-Guided Ternary Control by ablating the refinement weight $\beta$, with results shown in Table 2. The case where $\beta = 1.0$ is equivalent to using a standard binary mask, completely discarding the flow-based prior in masked regions. Conversely, $\beta = 0.0$ represents fully trusting the prior. Our proposed setting, $\beta = 0.5$, which empowers the model to *refine* the flow-warped prior, achieves the best performance across all reported metrics. This demonstrates that granting the model the flexibility to leverage the flow prior

| Weights | PSNR ↑ | SSIM ↑ | $E^*_{\text{warp}}$ ↓ | $\text{TC}_{\text{local}}$ ↑ |
|---|---|---|---|---|
| $\beta = 0.0$ | 34.32 | 0.9769 | 1.1163 | 0.9761 |
| $\beta = 0.25$ | 35.01 | 0.9811 | 1.1192 | 0.9758 |
| $\beta = 0.5$ | **35.84** | **0.9896** | **1.0997** | **0.9766** |
| $\beta = 0.75$ | 34.97 | 0.9847 | 1.1147 | 0.9749 |
| $\beta = 1.0$ | 34.66 | 0.9826 | 1.1116 | 0.9745 |

Table 2: Ablation study of the Ternary Mask refinement weight $\beta$. The setting $\beta = 1.0$ is equivalent to a binary mask. Our proposed setting, $\beta = 0.5$, achieves the best results across all metrics.

| Guidance | PSNR↑ | SSIM↑ | $E^*_{warp}$↓ | $\text{TC}_{\text{local}}$ ↑ |
|---|---|---|---|---|
| AR | 34.11 | 0.9744 | 1.1057 | **0.9767** |
| HR | 34.57 | 0.9788 | 1.1239 | 0.9747 |
| $\lambda = 0.05$ | 34.11 | 0.9735 | 1.1013 | 0.9767 |
| $\lambda = 0.1$ | 34.46 | 0.9802 | 1.1164 | 0.9757 |
| $\lambda = 0.15$ | 35.22 | 0.983 | 1.1021 | 0.9762 |
| $\lambda = 0.2$ | 34.53 | 0.9787 | **1.0995** | 0.9764 |
| $\lambda = 0.25$ | **35.84** | **0.9896** | 1.0997 | 0.9766 |
| $\lambda = 0.3$ | 34.63 | 0.9775 | 1.1004 | 0.9761 |
| $\lambda = 0.35$ | 33.52 | 0.9787 | 1.1152 | 0.9764 |

Table 3: Ablation study of our Adaptive Global-Local Guidance on the DAVIS dataset. We analyze the individual performance of AR and HR guidance and the effect of the motion sensitivity parameter $\lambda$.

while simultaneously correcting its inherent inaccuracies is critical for preserving high-frequency details and enhancing overall structural fidelity.

**Analysis of Adaptive Global-Local Guidance.** To validate our guidance strategy, we first isolate the effects of its constituent parts. Table 3 presents this analysis.

- Complementary Strengths of AR and HR: The first two rows compare using only Autoregressive (AR) guidance against only Hierarchical (HR) guidance. The AR-only model achieves the highest local temporal consistency ($\text{TC} = 0.9767$), confirming its strength in generating smooth frame-to-frame transitions. However, its PSNR and SSIM are lower than the HR-only model. Conversely, the HR-only model yields superior PSNR and SSIM scores, indicating better preservation of global structure and frame quality, but at the cost of slightly reduced local smoothness. This result empirically confirms our central hypothesis that AR and HR are complementary, motivating the need for their dynamic fusion.

- Impact of Motion-Adaptive Blending ($\lambda$): The subsequent rows in Table 3 demonstrate the effect of the motion sensitivity parameter $\lambda$, which controls the blend between AR and HR guidance. The results show a clear performance improvement as $\lambda$ increases from 0.05, with the optimal balance achieved at $\lambda = 0.25$. This setting yields the highest PSNR and SSIM scores while maintaining excellent temporal consistency. Performance begins to degrade beyond this point (e.g., $\lambda = 0.35$), suggesting that over-reliance on motion can be detrimental. This validates that our adaptive blending mechanism, which intelligently adjusts the guidance based on scene dynamics, is crucial for achieving state-of-the-art performance.

## 5 CONCLUSION

In this paper, we introduced *OmniPainter*, a novel video inpainting framework designed to resolve the fundamental conflict between global and local temporal consistency. Our approach is defined by two key innovations: a *Flow-Guided Ternary Control* for enhanced structural fidelity, and an *Adaptive Global-Local Guidance* strategy. This guidance mechanism dynamically blends specialist Autoregressive (AR) and Hierarchical (HR) scores, intelligently adapting to both scene motion and the current diffusion timestep to achieve a robust temporal equilibrium. Our extensive experiments demonstrate that OmniPainter establishes a new state-of-the-art by significantly mitigating both flickering and contextual drift, paving the way for more reliable and sophisticated video inpainting technologies.

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

# A APPENDIX

## A.1 LATENT UPSAMPLING

Most generative models struggle with high-resolution image generation, often causing structural distortions and quality loss. These issues are exacerbated when the output is generated at resolutions higher than those seen during training. To address this, *Upsample Guidance* Hwang et al. (2024) and *I-MAX* Du et al. (2024) first generate images at the native training resolution and then use them as guidance for producing higher resolution outputs, effectively improving stability and reducing artifacts. However, such methods require a full low-resolution generation path at every denoising step, which is computationally inefficient.

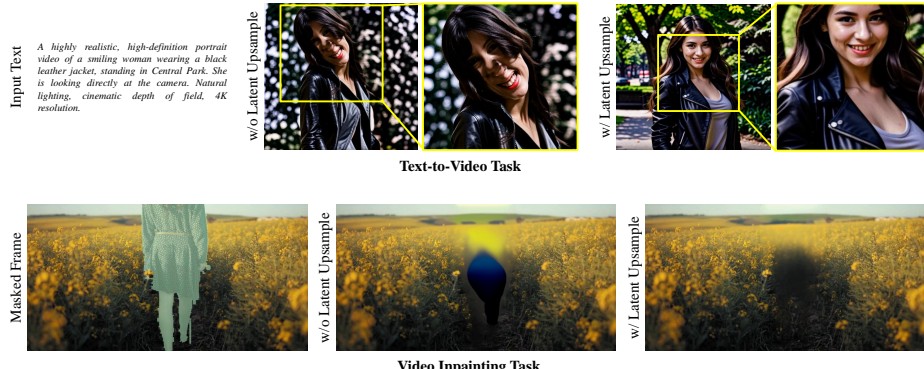

Figure 7: Qualitative results of latent upsampling. Top: text-to-video results at $1280 \times 1280$; Bottom: video inpainting results at $1920 \times 1080$ pixels. Both methods use models trained at a resolution of $512 \times 512$.

In contrast, we propose a simpler and more efficient strategy. Our method generates a low-resolution latent only during the early denoising stages, when the model primarily captures low-frequency structural information, and uses it to guide the subsequent high-resolution denoising. Furthermore, this method is applicable not only to video inpainting but also to other image and video generation tasks (see Figure 7).

Let $k$ be the total number of denoising timesteps, and define a threshold timestep $t_{\text{th}} = \tau \cdot k$ (with $\tau \in [0, 1]$) at which we switch from low-resolution to high-resolution denoising. The basic procedure is shown in Algorithm 1.

Here, $\mathcal{U}(\cdot)$ is a bilinear upsampling operator, while $\mathcal{E}(\cdot)$ and $\mathcal{D}(\cdot)$ denote the encoder and decoder of a pre-trained VAE, respectively. Since latent-space upsampling often results in blurriness, we use the encoder-decoder pair to perform upsampling in image space. Using the intermediate result $\mathbf{z}_0$ as a low-resolution guide at $t_{\text{th}}$, our approach effectively reduces the number of denoising steps

**Algorithm 1** Latent Upsampling

1: Set $t = k$
2: **while** $t \geq 0$ **do**
3:     **if** $t = t_{\text{th}}$ **then**
4:         $\bar{\mathbf{z}}_0 = \frac{1}{\sqrt{\bar{\alpha}_t}}(\mathbf{z}_t - \sqrt{1 - \bar{\alpha}_t}\epsilon_\theta(\mathbf{y}_t, t))$
5:         $\mathbf{z}_{t-1} = \sqrt{\bar{\alpha}_{t-1}}\,\mathcal{E}(\mathcal{U}(\mathcal{D}(\bar{\mathbf{z}}_0))) + \sqrt{1 - \bar{\alpha}_{t-1}}\,\epsilon$
6:     **else**
7:         $\mathbf{z}_{t-1} = \frac{1}{\sqrt{1-\beta_t}}(\mathbf{z}_t - \frac{\beta_t}{\sqrt{1-\bar{\alpha}}}\epsilon_\theta(\mathbf{y}_t, t)) + \sigma_t\,\epsilon$
8:     **end if**
9:     $t \leftarrow t - 1$
10: **end while**

Figure 8: Performance comparison across different latent upsampling hyperparameter settings.

compared to previous methods Hwang et al. (2024); Du et al. (2024), cutting the process from $2k$ to $k$ steps.

**Analysis of Latent Upsampling.** As illustrated in Figure 7(down), performing high-resolution inpainting directly without early latent upsampling yields artifacts and distorted pattern, as also reported in methods Hwang et al. (2024); Du et al. (2024). In contrast, our approach regulates the upsampling process by controlling the upsampling step through a parameter $\tau$. We conducted experiments using videos at $1920 \times 1080$p resolution from MiraDataset Ju et al. (2024) with random masks. Figure 8 presents the upsampling performance across different values of $\tau$, showing that the best results are obtained when $\tau = 0.25$.

## A.2 SELF-COLLECTED DATASET

We collected around 10k video clips, each ranging from 10 seconds to 2 minutes in duration. The dataset covers a wide range of domains, including variety shows, sports, dramas, movies, TV shows, and animations. To ensure video quality, we analyzed the aesthetic score Li et al. (2022a) of each clip and excluded those with low scores. We also calculated the average optical flow difference between adjacent frames Sun et al. (2018) to remove videos with minimal motion. To promote dataset diversity, we extracted CLIP Radford et al. (2021) features from sampled frames of each video and computed cosine similarity across all video pairs, excluding those with high similarity. As a result of this filtering process, we curated a final dataset of 2,036 high-quality and diverse videos, which were used for training *OmniPainter*.

## A.3 TRAINING DETAILS

**Training Configurations.** Our video inpainting latent diffusion model is trained using the Adam optimizer ($\beta_1 = 0.9, \beta_2 = 0.999$) with no weight decay. We use a fixed learning rate of $1 \times 10^{-5}$ without a learning rate scheduler. The diffusion process is configured with $T = 1000$ timesteps, following a linear noise schedule from $\beta_{\text{start}} = 10^{-4}$ to $\beta_{\text{end}} = 0.02$. Training is performed on 8 Nvidia A100 GPUs (80GB each). We set the per-GPU batch size to 1 and apply gradient accumulation by a factor of 4, resulting in an effective batch size of 64. The model is trained for a total of 50,000 iterations.

**Ternary Mask and augmentation for training.** We generate a **training ternary mask**, $m_t^{\text{train}}$, randomly for each ground-truth frame $x_t$. This mask is generated by creating random rectangular or free-form patches, ensuring the model learns spatially coherent inpainting and refinement. The pixels in $m_t^{\text{train}}$ are assigned one of three values based on predefined ratios:

- Known Regions ($\hat{m}_t(j) = 0$): A fraction of the pixels (e.g., 50%) are left untouched. The model learns from the original, clean data in these areas.

- Inpainting Regions ($\hat{m}_t(j) = 1$): Another fraction (e.g., 25%) is designated for a full inpainting task. These regions are completely masked out in the input.

- **Refinement Regions** ($\hat{m}_t(j) = \beta$): The remaining pixels (e.g., 25%) are assigned the value $\beta = 0.5$. These regions undergo a synthetic degradation process to mimic the imperfectly warped priors encountered during inference.

Specifically, the input $x'_t$ provided to the model during training is constructed as follows for each pixel $j$:

$$x'_t(j) = \begin{cases} x_t(j) & \text{if } m_t^{\text{train}}(j) = 0.0 \\ \mathcal{A}(x_t(j)) & \text{if } m_t^{\text{train}}(j) = 0.5 \\ \mathbf{c}_{\text{mask}} & \text{if } m_t^{\text{train}}(j) = 1.0 \end{cases} \tag{9}$$

where $\mathbf{c}_{\text{mask}}$ is a placeholder value (e.g., a gray color) for the inpainting regions. The function $\mathcal{A}(\cdot)$ represents our stochastic augmentation pipeline, which applies a random affine transformation followed by Gaussian noise to the original pixels $x_t(j)$.

Crucially, regardless of the input, the model is always tasked with reconstructing the original, uncorrupted frame $x_t$. This training strategy effectively teaches the model to perform three actions based on the mask value: copy known pixels (0.0), denoise and de-artifact corrupted priors (0.5), and missing content from scratch (1.0). This prepares the model to robustly handle the varied scenarios presented by the flow-guided ternary mask $\hat{m}_t$ during inference.

## A.4 ADDITIONAL QUALITATIVE RESULTS

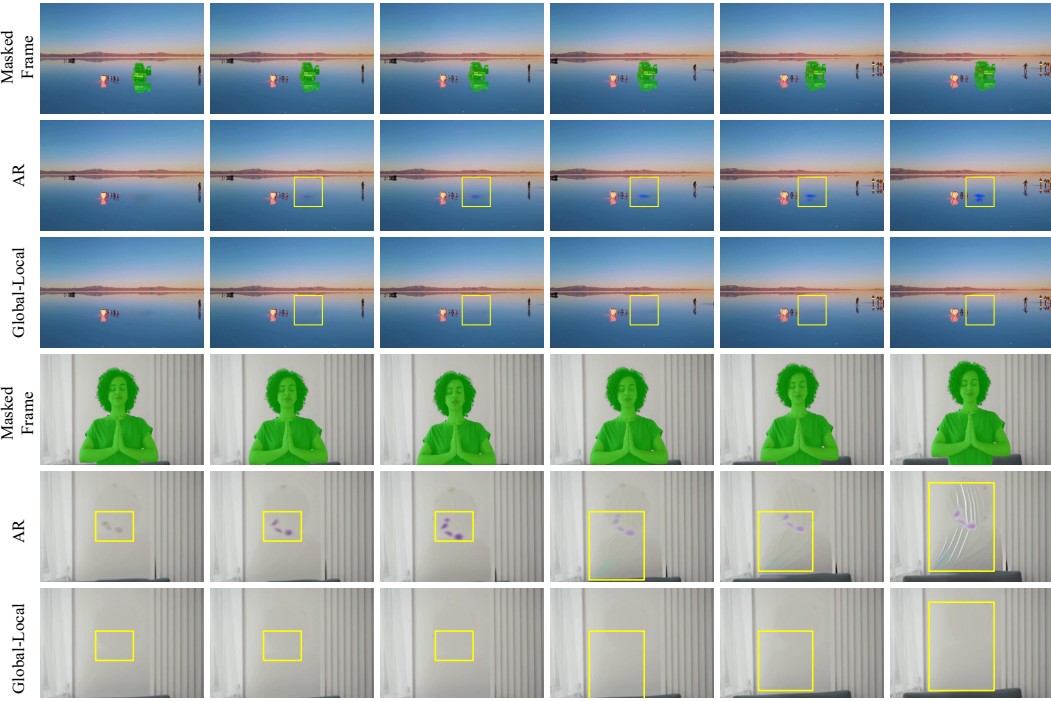

Figure 9: Qualitative Comparisons between AR and Global-Local guidance.

We present additional qualitative results demonstrating that our proposed approach significantly outperforms both the AR and HR methods.

Figure 9 illustrates that even minor errors in the AR approach tend to propagate over time, degrading the overall quality of the output, whereas our Global-Local guidance effectively prevents error accumulation, resulting in a stable and reliable performance even under challenging temporal variations.

As shown in Figure 10, while the AR approach suffers from noticeable inconsistencies in the inpainted context at the beginning and end of the sequence, our method consistently maintains natural, uniform textures throughout the entire sequence, ensuring robust global temporal consistency that is critical for long-term video restoration and generation tasks.

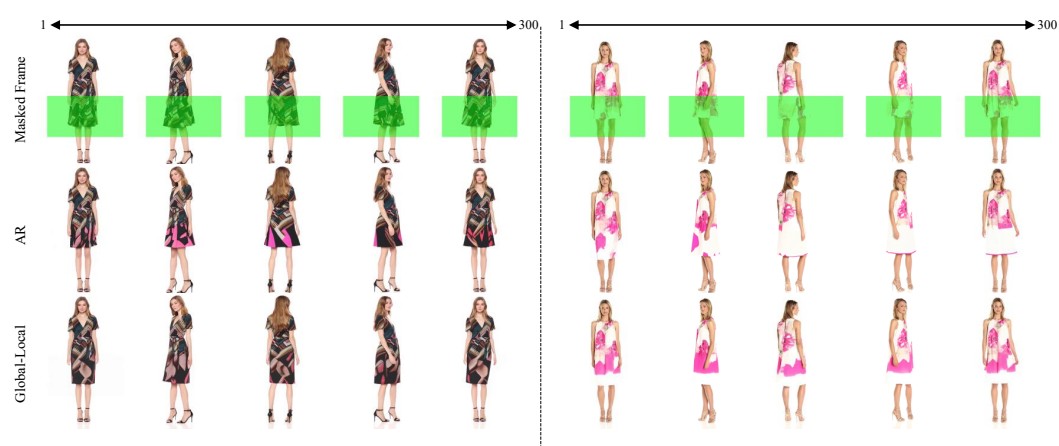

Figure 10: Qualitative Comparisons between AR and Global-Local guidance.

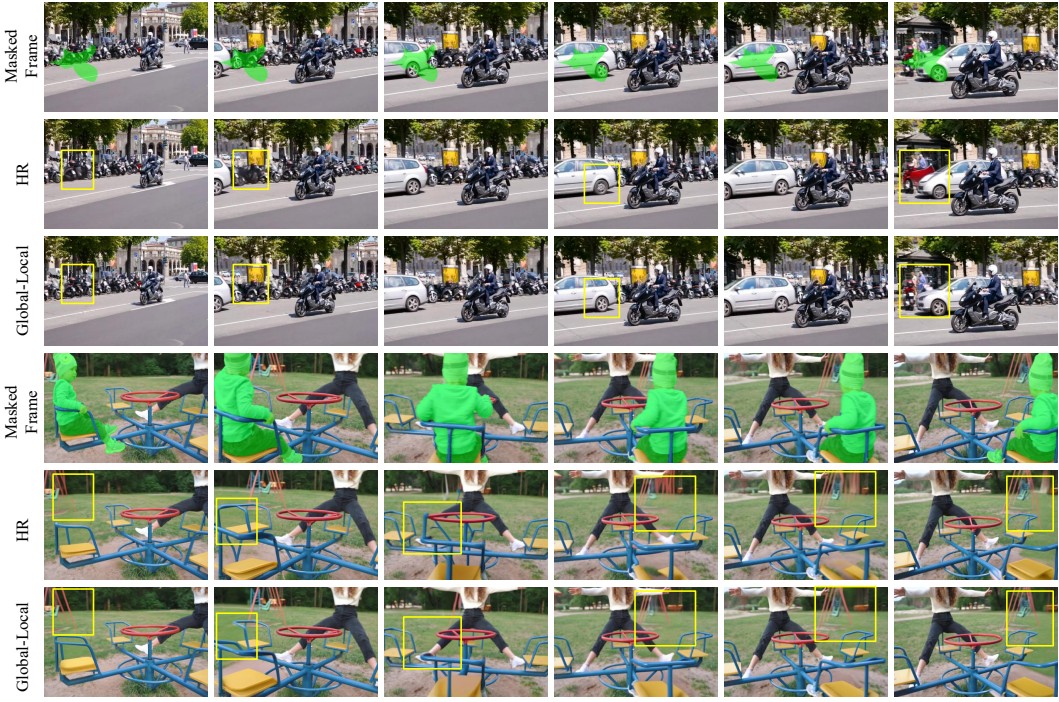

Figure 11: Qualitative Comparisons between HR and Global-Local guidance.

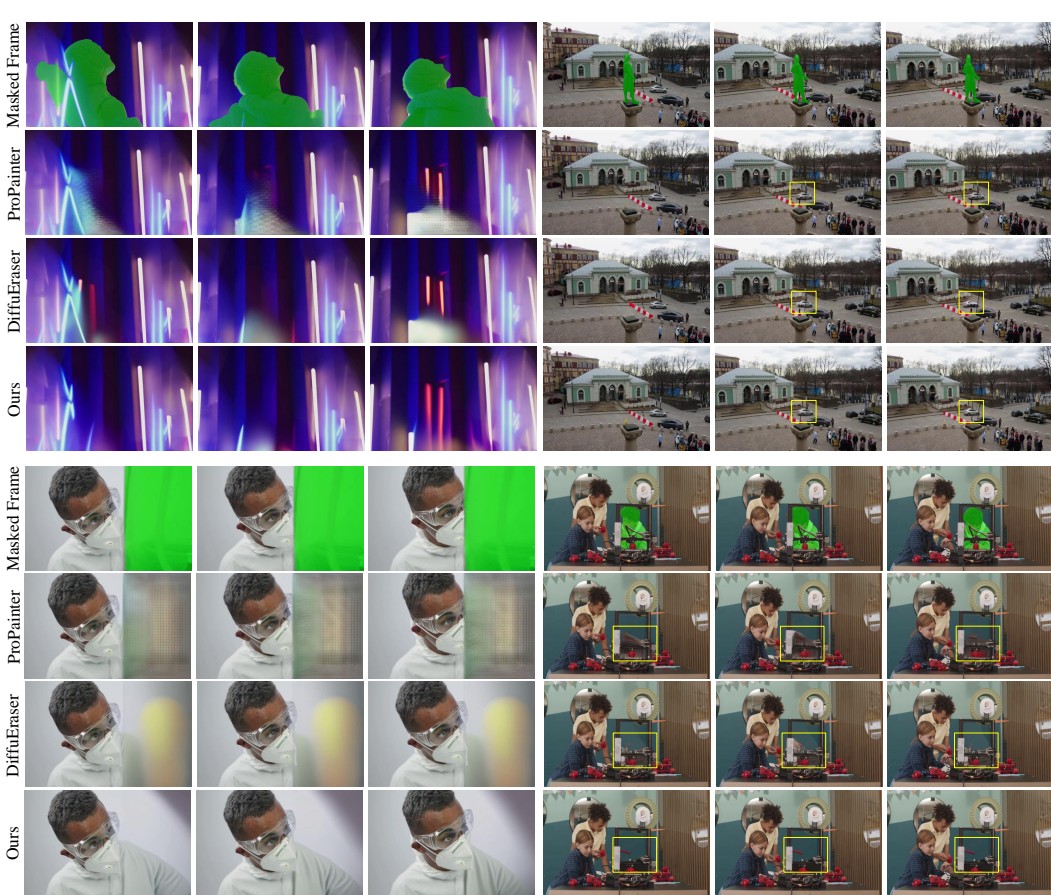

Figure 12: Additional Qualitative Comparisons.

Moreover, Figure 11 demonstrates that the HR approach experiences abrupt texture changes during fast motion, leading to local motion inconsistencies; in contrast, our approach delivers smooth, natural texture transitions that ensure motion coherence even in high-speed scenarios.

In addition, Fugure 12 shows qualitative comparisons with state-of-the-art methods such as DiffuEraser and ProPainter reveal that our technique consistently provides superior texture quality and enhanced temporal consistency, clearly overcoming the limitations of existing methods. Overall, our Global-Local guidance technique not only addresses the drawbacks inherent in AR and HR methods but also offers exceptional performance in preserving both global texture integrity and local motion smoothness, making it a highly effective solution for a wide range of video inpainting and restoration applications.

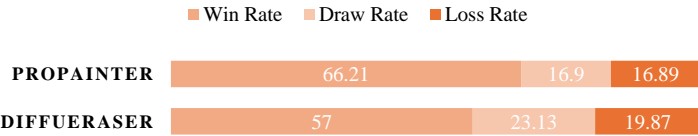

Figure 13: Human preference win rate.

## A.5 USER STUDY

To better compare the performance of different methods, we conducted a user study with 44 participants. The goal was to evaluate the visual quality of our method, *OmniPainter*, compared to two recent state-of-the-art approaches: *DiffuEraser* and *ProPainter*.

In each test, participants were shown two videos side by side: one generated by our method and the other by one of the baseline methods. They were asked to choose the video that looked better in terms of visual quality, or to say that both looked similar. This helped us to understand which method people preferred based on appearance.

We used 100 videos in total for the study, including 50 video completion examples and 50 object removal examples. All video pairs were performed using the same input masks for all methods to ensure a fair comparison.

The results are shown in Figure 13. Overall, users showed a clear preference for *OmniPainter* over both DiffuErase and ProPainter. This suggests that our method produces results that are more visually pleasing and consistent, confirming its effectiveness from a human perspective.

## A.6 COMPUTATIONAL RESOURCE ANALYSIS

All experiments for computational resource analysis were conducted on a machine equipped with an AMD EPYC 7502 32-Core CPU and a single NVIDIA A100 GPU with 80GB of memory. The reported computation time corresponds to processing 80-frame videos at a resolution of $512 \times 512$, averaged over 10 runs. Standard deviations are shown in parentheses. The measurement excludes any video pre-processing and post-processing steps. See Table 4 for details. For fair comparison, *ProPainter* and *DiffuEraser* were evaluated using their publicly released implementations with default settings provided by the authors.

|  | Inference time (seconds) | VRAM (MB) | NFE |
|---|---|---|---|
| Propainter | 23.44($\pm$0.73) | 15670 | 1 |
| Diffueraser | 35.29($\pm$0.34) | 15553 | 2 |
| Ours | 29.32($\pm$0.14) | 15670 | 8 |

Table 4: A comparison of computational resource consumption during inference. NFE refers to the number of function evaluations.

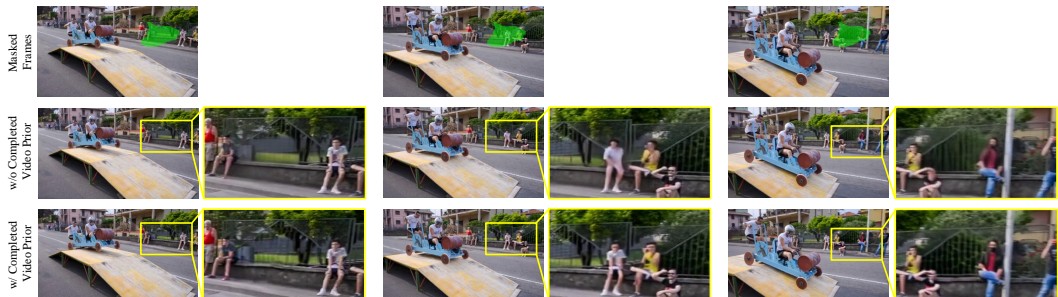

Figure 14: Qualitative comparison of the effect of flow-based completed video as a prior.

### A.7 ADDITIONAL ABLATION STUDY

**2. Ablation Study on Adaptive Blending Functions.** A core component of our Global-Local Guidance (GLG) is the adaptive blending of Autoregressive (AR) and Hierarchical (HR) scores. This blending is dynamically controlled by two weights: a motion-dependent weight, $\gamma$, and a denoising-step-dependent weight, $\eta$. To validate our design choices for the functions governing these weights, we conduct an ablation study, replacing our proposed functions with simpler or alternative formulations. The results, summarized in Table 5, demonstrate the superiority of our original design.

**On the Motion-Dependent Weight ($\gamma$).** Our proposed motion-dependent weight function, $\gamma = \Phi(x) = 1 - e^{-\lambda x}$, is designed to offer a nuanced response across a wide dynamic range of motion. To validate this design, we replace it with a simpler clamped linear function:

$$\gamma_{\text{Linear}} = \text{clamp}(a \cdot \mathcal{F}(X), 0, 1)$$

For this ablation, we test the linear model with different scaling constants, $a$.

As shown in Table 5, the linear models are markedly inferior. Their primary drawback is the **constant rate of response**. The weight $\gamma_{\text{Linear}}$ increases uniformly with the motion magnitude $\mathcal{F}(X)$ until it abruptly saturates. This makes the model difficult to tune: a small scaling factor $a$ leads to insensitivity for moderate motion, while a large $a$ causes premature saturation, failing to distinguish between slight and moderate motion intensities.

In contrast, our proposed exponential function $\Phi(x)$ provides more sophisticated control due to its **concave shape**. It exhibits high sensitivity to initial motion (i.e., the transition from static to dynamic), ensuring even slight movements begin to invoke AR guidance. However, its rate of increase diminishes as motion grows, allowing the weight to gracefully approach saturation. This non-linear behavior is crucial, as it allows the model to be responsive to low-to-moderate motion while remaining robust and not overreacting to extremely high motion values. This ensures that guidance is assigned appropriately based on the true intensity of the scene's dynamics.

**On the Denoising Schedule Weight ($\eta$)** Our scheduling weight, $\eta = \sigma(\log(\text{SNR}(k)))$, is designed to implement the "HR first" strategy by gradually shifting from HR to AR guidance as the signal-to-noise ratio increases. This function creates a smooth transition that is dynamically tied to the model's convergence state. We investigate whether this adaptive, signal-aware scheduling is truly necessary by comparing it against simpler schedules that depend only on the discrete timestep $k \in [1, K]$.

We test two alternatives:

1. **Linear Schedule**: A simple linear ramp, $\eta_{\text{Linear}} = k/K$.

2. **Exponential Schedule**: A schedule that starts very slowly and accelerates sharply, defined as $\eta_{\text{Exp}} = (\exp(1 - k/K))/(e - 1)$.

The results in Table 5 show that these predefined schedules fall short of our SNR-based approach. While the 'Linear $\eta$' and 'Exponential $\eta$' models provide a smooth transition, they are not adaptive to the actual denoising process. Our method, by linking $\eta$ to the SNR, allows the model to transition

from HR to AR guidance precisely when the signal becomes strong enough for local details to be refined. This confirms that the superiority of our "HR first" strategy stems not just from the sequence, but critically from the *smooth and adaptive* nature of the transition.

| Guidance Model | PSNR↑ | SSIM↑ | $E_{warp}^*$↓ | $\text{TC}_{\text{local}}$↑ |
|---|---|---|---|---|
| *Ablation on ($\gamma$)* | | | | |
| GLG w/ Linear $\gamma$ ($a = 0.025$) | 33.17 | 0.9715 | 1.1384 | 0.9610 |
| GLG w/ Linear $\gamma$ ($a = 0.05$) | 34.25 | 0.9758 | 1.1180 | 0.9742 |
| GLG w/ Linear $\gamma$ ($a = 0.1$) | 34.40 | 0.9765 | 1.1095 | 0.9750 |
| GLG w/ Linear $\gamma$ ($a = 0.15$) | 32.89 | 0.9582 | 1.571 | 0.9521 |
| GLG w/ Linear $\gamma$ ($a = 0.2$) | 31.11 | 0.9442 | 1.6112 | 0.9482 |
| *Ablation on ($\eta$)* | | | | |
| GLG w/ Linear $\eta$ | 35.15 | 0.9740 | 1.1290 | 0.9715 |
| GLG w/ Exponential $\eta$ | 34.28 | 0.9755 | 1.1220 | 0.9728 |
| **Ours (Full Model)** | **35.84** | **0.9896** | **1.0997** | **0.9766** |

Table 5: Ablation study of our Adaptive Blending Functions on the DAVIS dataset. We analyze the impact of replacing our proposed functions for $\gamma$ and $\eta$ with alternative formulations. Our full model (in bold) demonstrates superior performance across all variants.

## A.8 LIMITATIONS

While our proposed *OmniPainter* framework demonstrates strong performance in terms of visual quality and temporal consistency, several limitations remain, which point to meaningful directions for future research and development. Our approach assumes access to accurate and temporally stable segmentation masks. However, in real-world scenarios, such masks are typically generated using automated tools (e.g., SAM2), which may introduce inconsistencies or errors. These imperfections are not explicitly handled in the current framework and can lead to quality degradation. Improving robustness to imperfect mask inputs or integrating adaptive mask refinement modules could enhance the method's practical reliability. Addressing these limitations is essential for advancing video inpainting systems toward more efficient, autonomous, and deployment-ready solutions.

## A.9 IMPLEMENTATION OF WAN2.1-BASED ARCHITECTURE

To further validate the scalability of our OmniPainter framework, we explored its adaptation to the Wan2.1 Wang et al. (2025) video foundation model. While our primary experiments utilize Stable Diffusion v1.5 (SD1.5), integrating Wan2.1-1.3B necessitates addressing fundamental differences in both the diffusion objective and the latent space representation. This section details the specific architectural modifications required.

**Rectified Flow Objective.** The most significant divergence in the generative process lies in the training objective. Unlike SD1.5, which employs a standard $\epsilon$-prediction objective typical of Denoising Diffusion Probabilistic Models (DDPM), Wan2.1 adopts a **Rectified Flow** formulation. Consequently, the model is trained to predict the velocity field $v$ rather than the Gaussian noise $\epsilon$. The loss function is formulated as:

$$\mathcal{L}_{RF} = \mathbb{E}_{t,x_1,x_0} \left[ ||v_\theta(x_t, t) - (x_1 - x_0)||^2 \right] \tag{10}$$

where $x_1$ represents the clean video data, $x_0$ denotes the noise distribution, and $x_t$ is the interpolated state at timestep $t$. In inference, this requires replacing the DDIM scheduler with an Ordinary Differential Equation (ODE) solver compatible with flow matching.

**Spatio-Temporal VAE and Masking Strategy.** A critical challenge in adapting Wan2.1 is the distinct compression mechanism of its Variational Autoencoder (VAE).

- **SD1.5 VAE (Spatial Compression):** Encodes frames independently, mapping an input of shape $F \times H \times W$ to a latent feature map of $F \times \frac{H}{8} \times \frac{W}{8}$. This preserves the temporal dimension, allowing for precise frame-wise latent masking and slicing.

- **Wan2.1 VAE (Spatio-Temporal Compression):** Compresses both spatial and temporal dimensions. Specifically, it reduces the temporal axis by a factor of 4, resulting in a latent shape of $\frac{F}{4} \times \frac{H}{8} \times \frac{W}{8}$.

Due to this temporal downsampling, a single latent feature vector encodes information from 4 consecutive frames. Consequently, direct latent slicing to obtain specific conditioning frames (e.g., for Hierarchical or Autoregressive guidance) becomes infeasible, as a single latent index contains mixed information from adjacent frames.

**Pixel-Space Sampling** To resolve the ambiguity caused by temporal compression and ensure precise temporal alignment for both conditioning and consistency, we implement an explicit *Pixel-Space Sampling* strategy during the denoising sampling process.

Unlike standard LDM inpainting where conditioning latents are directly sliced from the intermediate state, we perform a *Decode-Sample-Encode* process. Let $\mathcal{D}_{wan}$ and $\mathcal{E}_{wan}$ denote the decoder and encoder of the Wan2.1 VAE. At each denoising step, given the estimated clean latent $\hat{y}_0$, we proceed as follows:

1. **Full Decoding:** We first project the estimated latent back to the pixel space to recover the temporal resolution:
$$\hat{X}_0 = \mathcal{D}_{wan}(\hat{y}_0) \tag{11}$$

2. **Pixel-Space Sampling for Conditioning:** To obtain accurate guidance conditions (e.g., $C_{HR}$ for Hierarchical or $C_{AR}$ for Autoregressive guidance), we sample the specific target frames directly from the decoded pixel volume $\hat{X}_0$. Let $P_{cond}$ be the set of sampled pixel frames. We then re-encode them to obtain the aligned latent condition:
$$C_{cond} = \mathcal{E}_{wan}(P_{cond}), \quad \text{where } P_{cond} \subset \hat{X}_0 \tag{12}$$

This ensures that the LDM receives temporally precise conditions free from compression artifacts.

Although this strategy introduces additional computational overhead due to the repeated VAE passes, it is essential for handling the 3D-compressed latent space of Wan2.1. It ensures that both the conditioning frames (used for Global-Local Guidance) and the unmasked regions remain temporally consistent and aligned throughout the generation process.

### A.10 COMPARISONS WITH DiT-BASED METHODS

To demonstrate the scalability and effectiveness of our proposed framework, we conducted a comprehensive comparison with recent state-of-the-art methods based on Diffusion Transformers (DiT). Specifically, we compare our **OmniPainter (Wan2.1)** implementation against two representative baselines:

- **MiniMax-Remover** Zi et al. (2025): A commercial video inpainting tool based on the Wan2.1-1.3B architecture.
- **VideoPainter** Bian et al. (2025): A video editing model based on CogVideoX-5B Yang et al. (2024), which employs an autoregressive strategy for long video generation.

All baseline methods were evaluated using their official default configurations.

**Quantitative Comparison.** Table 6 presents the quantitative results on the DAVIS and UBC Fashion datasets. OmniPainter (Wan2.1) outperforms both MiniMax-Remover and VideoPainter across all metrics. Notably, on the UBC Fashion dataset, which evaluates long-term consistency, our method achieves a $TC_{global}$ score of **0.9873**, significantly surpassing VideoPainter (0.1259). This stark difference quantifies the severe context drift observed in VideoPainter's autoregressive approach. Furthermore, our Wan2.1-based implementation shows a clear improvement over our SD1.5-based version, validating the benefits of scaling to stronger foundation models.

**Qualitative Analysis.** While MiniMax-Remover delivers competitive visual quality on short sequences, its performance degrades noticeably as video length increases. This limitation stems from

| Methods | DAVIS | | | | | UBC Fashion | | |
|---|---|---|---|---|---|---|---|---|
| | PSNR↑ | SSIM↑ | VFID↓ | $E_{warp}$↓ | TC$_{local}$↑ | VFID↓ | TC$_{local}$↑ | TC$_{global}$↑ |
| MiniMax-Remover | 34.10 | 0.9789 | 0.889 | 1.087 | 0.9876 | 0.279 | 0.9765 | 0.9795 |
| VideoPainter | 33.48 | 0.9658 | 0.1789 | 1.158 | 0.9857 | 0.348 | 0.9625 | 0.1259 |
| Ours (SD 1.5) | 35.84 | 0.9896 | 0.088 | 1.0997 | 0.9766 | 0.042 | 0.9845 | 0.9861 |
| **Ours (Wan2.1)** | **36.41** | **0.9911** | **0.068** | **0.934** | **0.9970** | **0.032** | **0.9886** | **0.9873** |

Table 6: Quantitative comparison with DiT-based state-of-the-art methods on DAVIS and UBC Fashion datasets. We evaluate MiniMax-Remover (Wan2.1-1.3B based) and VideoPainter (CogVideoX-5B based) using their default settings. The best results are marked in bold.

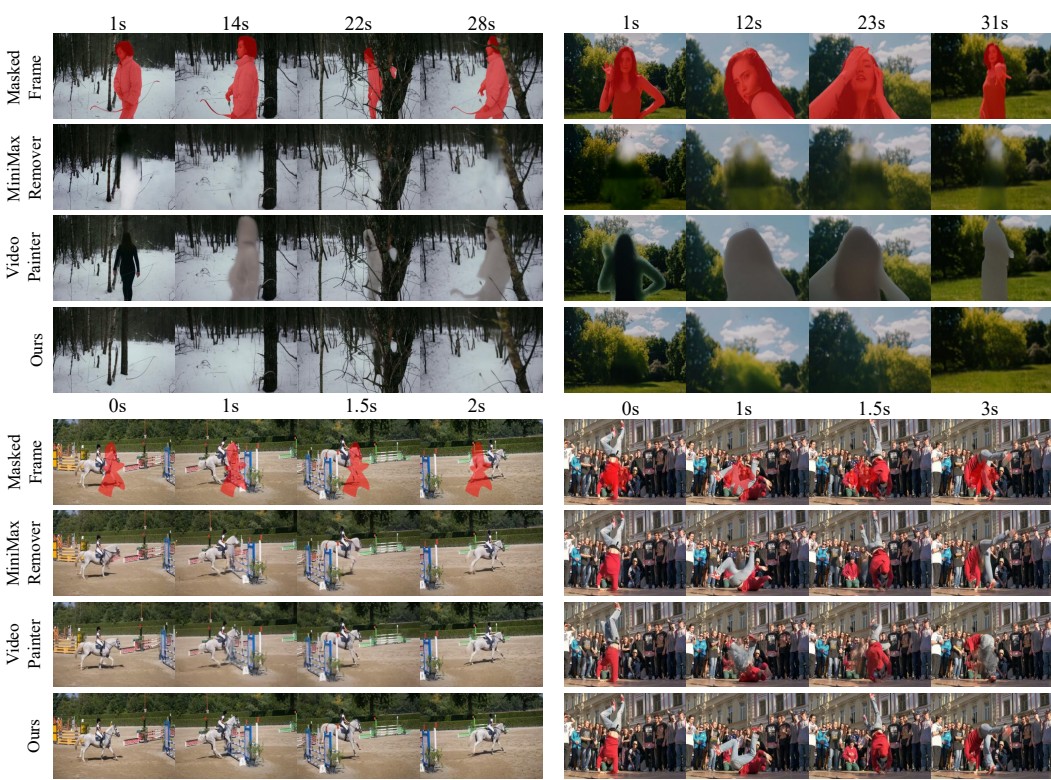

Figure 15: Qualitative comparison against DiT-based methods: MiniMax-Remover and VideoPainter.

the absence of a dedicated mechanism for modeling long-range dependencies, causing the model to struggle with maintaining global coherence over time.

VideoPainter, being primarily designed for instruction-based video editing, exhibits significant limitations in object removal tasks. It is highly sensitive to mask shapes and text prompts, often failing to cleanly remove objects or generating hallucinations. Moreover, as it relies on a standard autoregressive inference strategy for long videos, it suffers from severe error propagation; minor artifacts in early frames accumulate rapidly, leading to complete structural collapse in later frames.

In contrast, OmniPainter effectively addresses these issues through our proposed Adaptive Global-Local Guidance. By dynamically balancing short-term smoothness and long-term structure, our method remains robust against error accumulation and maintains high fidelity even in extended sequences.

**Efficiency Analysis.** We further evaluated the computational efficiency of each method (see Figure 17). Our analysis reveals that OmniPainter is the most efficient in terms of both inference speed and VRAM consumption.

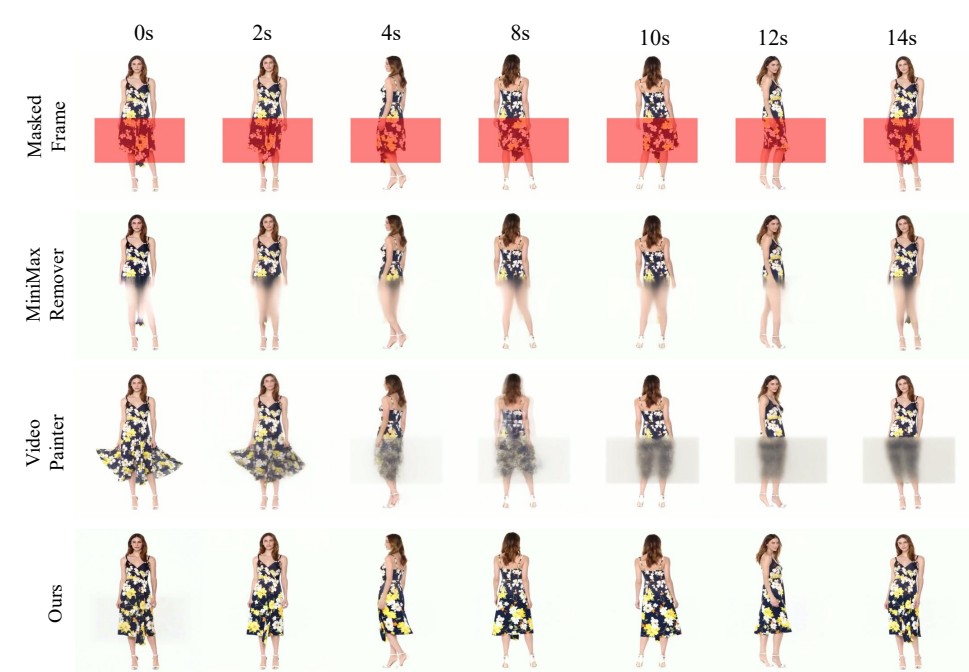

Figure 16: Qualitative comparison against DiT-based methods: MiniMax-Remover and VideoPainter.

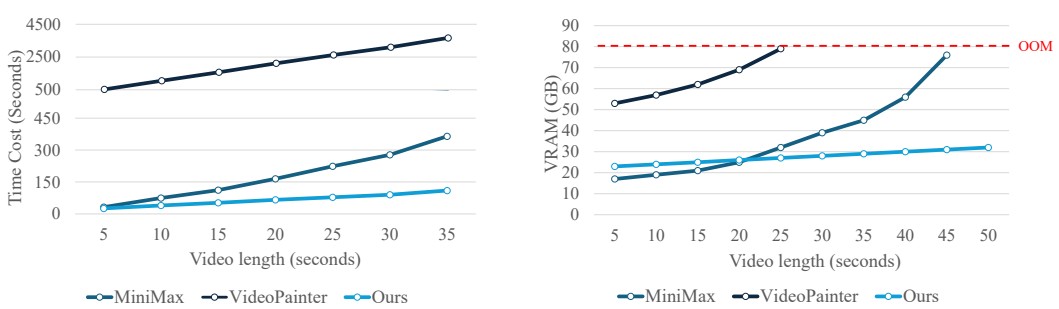

Figure 17: Comparison of inference time and VRAM efficiency across varying video lengths.

- **MiniMax-Remover:** While its speed is comparable to ours for short clips (6 NFE), it slows down significantly as video length increases. This is because it processes a growing number of temporal tokens simultaneously without an efficient windowing strategy, leading to a non-linear increase in computational cost and high VRAM usage.

- **VideoPainter:** Although its time cost increases linearly, the baseline overhead is extremely high due to the requirement of 50 NFE and the heavy computational load of the CogVideoX-5B model.

- **Ours:** By dividing the video into manageable groups and employing our efficient guidance strategy, our method achieves a linear increase in processing time with a low constant factor, maintaining consistently low VRAM usage regardless of video length.

## A.11 OVERALL PIPELINE PSEUDOCODE

**Algorithm 2** OmniPainter Full Inference Pipeline

**Require:** Masked Video $X$, Binary Mask $M$, Model Architecture $\mathcal{A}$ (e.g., Wan2.1 or SD1.5)
**Ensure:** Inpainted Video $\tilde{X}$

    *Stage 1: Flow-Guided Preprocessing (Ternary Control)*
1: **for** each frame $t$ in Video **do**
2:     $F, A_r \leftarrow$ **Estimate Optical Flow** between $x$ and $x_t$
3:     $\hat{x}_t \leftarrow$ **Warp** $x$ using $F$ to pre-fill masked regions
4:     $\hat{m}_t \leftarrow$ **Create Ternary Mask** using $M$ and $A_r$
5:     $z_t \leftarrow$ **VAE Encode** pre-filled frame $\hat{x}_t$
6: **end for**
7: $\mathcal{F}(X) \leftarrow$ Calculate Global Motion Score from all flows

    *Stage 2: Adaptive Global-Local Denoising*
8: $y_K \leftarrow$ Initialize Gaussian Noise Latents
9: **for** step $k = K$ to $1$ **do**
10:     $\omega \leftarrow$ **Compute Adaptive Weight**$(\mathcal{F}(X)$, step $k)$
11:     $\hat{y}_0 \leftarrow$ **Estimate Clean Latent** from noisy $y_k$
12:     **if** $\mathcal{A}$ is Wan2.1 **then**
13:         $\hat{X}_0 \leftarrow \mathcal{D}(\hat{y}_0)$         ▷ *Decode latent to pixel space first*
14:     **end if**
15:     **for** each frame $t$ **do**
16:         **// 1. Hierarchical (Global) Guidance**
17:         **if** $\mathcal{A}$ is Wan2.1 **then**
18:             $P_{HR} \leftarrow$ Sample sparse **Keyframes** from $\hat{X}_0$     ▷ *Sample pixels*
19:             $C_{HR} \leftarrow \mathcal{E}(P_{HR})$     ▷ *Encode pixel samples*
20:         **else**
21:             $C_{HR} \leftarrow$ Sample sparse **Keyframes** from $\hat{y}_0$     ▷ *Sample latent directly*
22:         **end if**
23:         $\epsilon_{HR} \leftarrow$ **LDM**$(y_t^k, k, C_{HR}, z_t, \hat{m}_t)$
24:         **// 2. Autoregressive (Local) Guidance**
25:         **if** $\mathcal{A}$ is Wan2.1 **then**
26:             $P_{AR} \leftarrow$ Sample **Previous Group** frames from $\hat{X}_0$     ▷ *Sample pixels*
27:             $C_{AR} \leftarrow \mathcal{E}(P_{AR})$     ▷ *Encode pixel samples*
28:         **else**
29:             $C_{AR} \leftarrow$ Sample **Previous Group** frames from $\hat{y}_0$     ▷ *Sample latent directly*
30:         **end if**
31:         $\epsilon_{AR} \leftarrow$ **LDM**$(y_t^k, k, C_{AR}, z_t, \hat{m}_t)$
32:         **// 3. Update Latent**
33:         $\epsilon_{final} \leftarrow$ **Blend Scores**$(\epsilon_{HR}, \epsilon_{AR}, \omega)$
34:         $y_t^{k-1} \leftarrow$ **Denoising Step**$(y_t^k, \epsilon_{final})$
35:     **end for**
36: **end for**

    *Stage 3: Final Reconstruction*
37: $\tilde{X} \leftarrow$ **VAE Decode** final latents $y_0$
38: **return** $\tilde{X}$

