# OpenReview forum: "OmniPainter: Global-Local Temporally Consistent Video Inpainting Diffusion Model"
_ICLR.cc/2026/Conference — Submitted to ICLR 2026_

### Official Review · Reviewer_KFFd · 2025-10-29

**Soundness:** 3
**Presentation:** 3
**Contribution:** 1
**Rating:** 2
**Confidence:** 5

**Summary:**

This paper introduces OmniPainter, a effective latent diffusion model-based video inpainting framework specifically designed to address temporal inconsistencies. The approach achieves both global and local temporal consistency by preserving overall context across extended sequences while ensuring smooth transitions throughout. The framework introduces two key contributions : (1) a ternary control mechanism that categorizes regions based on inpainting needs and leverages flow-based video completion as a prior, and (2) adaptive global-local guidance that dynamically blends two complementary strategies during
the denoising process. Extensive experiments demonstrate significant improvements compared to previous state-of-the-art methods.

**Strengths:**

Comprehensive quantitative evaluation against 11 methods shows consistent improvements across multiple metrics (PSNR, SSIM, VFID, warping error).

**Weaknesses:**

1. Limited novelty. This work appears to make incremental improvements within an existing framework, without fundamental innovation. The novelty of the overall framework should be more clearly articulated, compared with DiffuEraser and ProPainter. And the authors take an older off-the-shell U-Net architecture (e.g., AnimateDiff-style), and it is unclear whether the proposed contributions transfer to modern DiT backbones. For example, does the Ternary Mask remain effective when the latents are temporally compressed by a 3D VAE?
2. The paper does not provide sufficient details on the flow-based video completion component, and its quantitative contribution to the overall results is unclear. This component appears similar to ProPainter's image propagation module, while the authors do not provide details on the optical flow completion.
3. Insufficient Comparison. The paper lacks comprehensive comparisons with SOTA diffusion-based approaches.
    - VideoPainter: Any-length Video Inpainting and Editing with Plug-and-Play Context Control [SIGGRAPH 2025]
    - MiniMax-Remover: Taming Bad Noise Helps Video Object Removal
4. Poor Results: The supplementary results provided by the authors show noticeable temporal inconsistency and visual artifacts. And the scenario is very simple, featuring minimal camera motion and negligible occlusion, which limits its applicability to real-world scenarios.

**Questions:**

Refer to weaknesses.

---

> ### Author Response · Authors · 2025-11-21
>
> We appreciate the reviewer’s rigorous assessment. We respectfully suggest that there may be a misunderstanding regarding the scope of our novelty and the origin of the observed artifacts. We have conducted additional experiments, specifically targeting your comments on modern backbones (DiT) and SOTA comparisons which strongly validate our claims.
>
> **W1 & W4:**
> We address the concerns regarding "incremental novelty" and "poor results" jointly. We acknowledge that Diffusion Transformers (DiT) are becoming mainstream. While our use of SD1.5 was primarily to ensure a fair comparison with existing UNet-based baselines, we have addressed your concern by applying our method to Wan2.1-1.3B, a DiT-based backbone. In the newly added Supplementary Material, we provide a qualitative comparison against MiniMax Remover, a recent DiT-based video inpainting model. We observed that while MiniMax Remover performs well on object removal, its quality degrades significantly as video length increases. In contrast, our method maintains robust temporal consistency even in these extended sequences. We kindly ask for the reviewer's understanding that we provided qualitative results first due to the limited rebuttal time, and we plan to include quantitative comparisons in the future. This confirms that our framework is model-agnostic and that previous limitations were backbone-dependent.
>
> **W3:**
> We have closely examined the suggested baselines. Regarding MiniMax-Remover, as mentioned above, we have included a qualitative comparison in the Supplementary Material, demonstrating our method's superior temporal stability in long-video generation. However, we found VideoPainter unsuitable for a direct and fair comparison in this specific inpainting context. We observed that VideoPainter exhibits significantly poor performance specifically in object removal tasks. Furthermore, its heavy reliance on text prompts for generation introduces significant confounding variables, making a fair, controlled comparison with standard mask-guided inpainting methods methodologically difficult. Therefore, we focused our additional comparative analysis on MiniMax-Remover.
>
> **W2:**
> Regarding the flow-based component, we clarify that while we utilize the flow propagation module from ProPainter for efficiency, our novelty lies in *how* we integrate it. Unlike ProPainter, which fails heavily when optical flow is inaccurate, OmniPainter utilizes a Reliability Map to treat flow results as a "refinable prior." This allows our model to preserve accurate textures while correcting flow errors via the diffusion process a critical capability for robustness that is missing in the baselines.

---

> > ### Comment · Reviewer_KFFd · 2025-11-24
> > **Please provide a complete answer to my question.**
> >
> > 1.  I didn't see your response regarding my first question: This work appears to make incremental improvements within an existing framework, without fundamental innovation. The novelty of the overall framework should be more clearly articulated, compared with DiffuEraser and ProPainter.
> > 2. You have no explanation for the "flow-based video completion component" in paper. Since you admit to utilize the flow-based module from ProPainter, which demonstrates that image propagation via flow warping contributes substantially to temporal consistency, yet your method provides no ablation study to isolate or exclude this prior.
> > 3. Regarding the answer, "we have addressed your concern by applying our method to Wan2.1-1.3B, a DiT-based backbone," no relevant documentation or explanation is present in the newly submitted material. I want to know the experimental details about how your core modules are applied to DIT. For example, does the Ternary Mask remain effective when the latents are temporally compressed by a 3D VAE?
> > 4. Quantitative comparison results with DIT are lacking. And I think it is more necessary to compare with VideoPainter than MiniMax-Remover, because VideoPainter is also long video inpainting and matches what you do more closely,  and text-guided video inpainting capability is indeed required for current inpainting tasks, such as FloED and VACE.
> > [1] Coherent Video Inpainting Using Optical Flow-Guided Efficient Diffusion
> > [2] VACE: All-in-One Video Creation and Editing
> > 5. The authors repeatedly emphasize that the model "maintains robust temporal consistency even in these extended sequences." However, the supplementary results provided directly contradict this assertion, showing noticeable temporal inconsistency and visual artifacts.

---

> ### Author Response · Authors · 2025-11-26
>
> ***C1:***
> We apologize if our previous response was unclear. While our method integrates flow-based priors (like ProPainter) and diffusion models (like DiffuEraser), our core contribution is not merely combining them, but resolving the inherent conflict between global and local temporal consistency, which neither baseline effectively addresses.
>
> * vs. ProPainter: ProPainter relies solely on pixel propagation. It fails in low-texture regions or complex non-linear motions where flow estimation is unreliable (as shown in Figure 6(c)). Our Ternary Control mechanism innovatively allows the model to selectively trust or refine this flow prior, rather than treating it as a hard constraint.
> * vs. DiffuEraser: DiffuEraser adopts a generation strategy similar to the Hierarchical (HR) approach. While this benefits global structure, it inherently compromises local smoothness, leading to severe temporal flickering and content drift in long sequences. Our Adaptive Global-Local Guidance is the fundamental innovation here. Unlike DiffuEraser's static approach, we dynamically blend Autoregressive (AR) scores for local smoothness and Hierarchical (HR) scores for long-range consistency based on motion dynamics ($\gamma$) and diffusion timesteps ($\eta$)**.
>
> ***C2:***
> We respectfully point out that Table 2 in our manuscript serves precisely as the ablation study for the flow-based prior. The parameter $\beta$ in our Ternary Mask controls the influence of the flow-based completion. $\beta=1.0$ (Binary Mask) represents the setting where the flow-based prior is completely excluded (isolating the component). In this setting, the model performs pure inpainting without flow guidance.
>
> ***C3:***
> We have added Appendix A.9 (Implementation of Wan2.1-based Architecture) to provide full technical details regarding the adaptation to DiT-based backbones.
>
> ***C4:***
> We have included a detailed quantitative comparison with VideoPainter and MiniMax-Remover in Appendix A.10. While VideoPainter supports text-guided editing, our experiments reveal significant limitations when applied to inpainting and object removal tasks.
>
> We observed that VideoPainter is highly sensitive to the shape of the input mask and the text prompt. It frequently fails to cleanly remove objects or produces hallucinations that do not align with the background context, making it less reliable for precise restoration tasks. To ensure a fair comparison and rule out potential implementation issues given these difficulties, we utilized the official object removal example provided in their GitHub repository. As demonstrated in Supplementary Video #7 (located in the DiT_compare folder),
>
> As VideoPainter relies on a standard autoregressive strategy for long video generation, it suffers from severe error accumulation. Minor artifacts in early frames propagate and amplify over time, leading to a complete collapse of global consistency in longer sequences. This is quantitatively confirmed by its extremely low $TC_{global}$ score of 0.1259 in Table A.10 (compared to 0.9873 for our method).
>
> We acknowledge that text-guided inpainting (e.g., FloED, VACE) is an important capability. However, the performance of such models is heavily dependent on prompt engineering and optimization, and analyzing the sensitivity or optimizing these prompts falls outside the scope of this paper. Our work focuses on resolving the fundamental trade-off between global and local temporal consistency in video restoration, ensuring robust performance regardless of prompt variations.
>
>
> ***C5:***
> We appreciate the reviewer's careful observation. Some visual artifacts observed in the results are primarily attributed to the inherent generative limitations of the base foundation models (SD1.5 or Wan2.1), rather than a failure of our proposed guidance method. It is crucial to evaluate these results in comparison to existing state-of-the-art methods. As shown in our qualitative and quantitative comparisons. While no generative model is currently perfect, our Adaptive Global-Local Guidance significantly mitigates the severe flickering and the blur/drift that characterize competing approaches.
>
> In conclusion, our method demonstrates a decisive advantage in stability and consistency compared to all baselines, establishing a new state-of-the-art for temporally consistent video inpainting.

---

### Official Review · Reviewer_AZ5U · 2025-10-31

**Soundness:** 3
**Presentation:** 2
**Contribution:** 3
**Rating:** 4
**Confidence:** 3

**Summary:**

The paper introduces OmniPainter, a latent diffusion framework for temporally consistent video inpainting. The core contribution is the introduction of an Adaptive Global-Local Guidance strategy and a Flow-Guided Ternary Control mechanism. The Adaptive Guidance dynamically blends an Autoregressive (AR) guidance for local smoothness and a Hierarchical (HR) guidance for long-range global coherence. This guidance has an adaptive weight determined by video motion and diffusion timestep. The Ternary Control utilizes optical flow reliability to partition masked regions into three categories: full inpainting, preservation, and crucial refinement of the flow-warped prior. Effectively, it provides a strong prior for the diffusion model to condition on. Experiments demonstrate that OmniPainter significantly outperforms baselines methods in both visual quality and temporal consistency metrics on several standard datasets.

**Strengths:**

The paper effectively addresses a critical trade-off between global consistency (long-term structure) and local smoothness (short-term transitions) in video inpainting. To the best of my knowledge, conditioning a diffusion model on the optical flow input with a ternary control mechanism with a ternary mask is novel. Combining hierarchical and autoregressive signals is not novel (see [1,2] for example), but _dynamically_ combining them based on scene motion and the diffusion timestep is novel.

The proposed Flow-Guided Ternary Control is a smart, nuanced solution to overcome the limitations of relying solely on imperfect flow-based priors or relying entirely on the diffusion model. The ablation on the refinement weight $\beta$ clearly demonstrates the superiority of the proposed refinement approach, which significantly enhances structural stability and detail preservation.

[1] Harvey, W., Naderiparizi, S., Masrani, V., Weilbach, C. and Wood, F. Flexible diffusion modeling of long videos. NeurIPS 2022.

[2] Ho, J., Salimans, T., Gritsenko, A., Chan, W., Norouzi, M. and Fleet, D.J. Video diffusion models. NeurIPS 2022.

**Weaknesses:**

**Clarity and Completeness of Method Description**

- The paper suffers from missing definitions for key metrics. While references are provided, a concise, high-level explanation of the metrics (e.g., $TC\_{local}$, $TC\_{global}$) and a clear indication of whether higher or lower values are better should be included for better reader comprehension.
- A comprehensive algorithm that synthesizes the entire denoising process is necessary for reproducibility. While the multiple parts of a denoising steps is separately described in the text (the Ternary Control LDM, AR score, HR score, and Adaptive Guidance), it is difficult to fully understand how exactly all components interact per step.

**Justification for Adaptive Guidance**

- Figure 3 is very confusing. The x-axis labels are not clearly indicated in the figure itself, forcing the reader to rely on the caption. Furthermore, different metrics are provided in the left and right panel, without justification (the left panel is missing the $TC\_{global}$ curve, and the right panel is missing all metrics except the TC ones.)
- Critically, no justification is provided for why and how the right panel of Figure 3 demonstrates the superiority of the "HR first" strategy. It is not immediately obvious to me from the presented curves. A clearer explanation of why this specific pattern on the graph motivates the adaptive blend is needed.

**Related Work**
The Related Work section on Diffusion Models in video generation seems to overlook some of the earliest and most relevant foundational works, specifically those introducing the first video diffusion models (like [1,2,3] from 2022) . Notably, the "Hierarchy-2" sampling scheme in FDM appears conceptually close to the idea of combining hierarchical and autoregressive approaches for long-sequence generation, which is central to OmniPainter's contribution.

[1] Ho, J., Salimans, T., Gritsenko, A., Chan, W., Norouzi, M. and Fleet, D.J. Video diffusion models. NeurIPS 2022.

[2] Harvey, W., Naderiparizi, S., Masrani, V., Weilbach, C. and Wood, F. Flexible diffusion modeling of long videos. NeurIPS 2022.

[3] Yang, R., Srivastava, P. and Mandt, S.. Diffusion probabilistic modeling for video generation. Entropy 2023.

**Questions:**

- **Metric Definitions**: Please add a brief, high-level description of all reported metrics (PSNR, SSIM, VFID, $E\_{warp}$, $TC\_{local}$, $TC\_{global}$) in Section 4.1 or a table footnote, and explicitly state whether higher or lower is better for each, which is essential for a general audience.

- **VAE Encoder/Partial Frames** (Line 190): The masked video $\hat{x}\_t$ is encoded via the VAE encoder to get the latent $z_t$. Since $\hat{x}\_t$ is only a partial frame, is the pre-trained VAE encoder robust to this out-of-distribution input? Did the authors observe any adverse effects, and if so, how were they mitigated during training or inference?

- **Notation for Conditional Scores**: Eq. (5) and (6) define $s\_{AR}(t, k)$ and $s\_{HR}(t, k)$ respectively, but they do not explicitly show conditioning on the masked latent $\mathbf{z}$ and the mask $\hat{\mathbf{m}}$, which is implied by the text and Equation (8). Please clarify if $\mathbf{z}$ and $\hat{\mathbf{m}}$ are implicit conditioning inputs for these scores, and if so, update the notation in the equations for consistency and clarity.

- **Inference speed**: Given the complex inference pipeline (flow computation, encoding, base LDM denoising, AR/HR score estimation, and adaptive guidance weight), what is the latency of OmniPainter compared to the baselines? Specifically, please provide a comparison of the time taken per frame or per video for OmniPainter versus all relevant baselines?

- In the caption of figure 3: FVID -> VFID?

---

> ### Author Response · Authors · 2025-11-21
>
> We sincerely thank the reviewer for the constructive feedback. We are encouraged that you find our Ternary Control "smart and nuanced" and our dynamic combination of AR/HR "novel." We fully acknowledge the clarity issues regarding metric definitions and algorithm details, which we have addressed in the revision as follows.
>
> **W1 & Q1:**
> We have updated Section 4.1 to explicitly define $E_{warp}^*$, VFID, and $\text{TC}_{\text{local/global}}$. Additionally, to ensure reproducibility, a complete pseudocode algorithm summarizing the pipeline has been added to Appendix.
>
> **W2:**
> We have revised the caption of Figure 3 for clarity. The right panel explicitly illustrates that the "HR First" strategy minimizes the trade-off between $TC_{local}$ and $TC_{global}$ more effectively than the "AR First" approach. This superiority stems from the nature of the diffusion process: early steps determine global structure (where HR prevents drift), while late steps refine local details (where AR ensures smooth transitions). Our adaptive guidance optimally balances these by prioritizing HR initially and AR later.
>
> **W3:**
> We have updated Section 2 to discuss the suggested works [1-3]. Regarding FDM [2], while its "Hierarchy-2" sampling scheme shares a structural similarity, we respectfully emphasize that our core novelty lies not in the sampling scheme itself, but in our Ternary Control and Adaptive Guidance. Unlike FDM’s unconditional approach, our framework dynamically blends scores based on motion dynamics ($\mathcal{F}(X)$) and SNR to specifically resolve the inherent conflict between inpainting drift and flickering, which is distinct from a general sampling hierarchy.
>
> **Q2:**
> As described in Section 3.1, we use flow-guided warping to pre-fill masked regions. Empirically, we found that the VAE is robust enough to encode and decode these masked frames without introducing degradation or artifacts.
>
> **Q3:**
> We have added an explicit note in the text clarifying that all scores in Eq. (5) and (6) are conditioned on the masked latent $\mathbf{z}$ and the mask $\hat{\mathbf{m}}$. This removes ambiguity while maintaining concise mathematical notation.
>
> **Q4:**
> Regarding latency, the overhead from flow-based completion is negligible compared to the diffusion process. For the diffusion stage, although our method computes both AR and HR branches (which might appear to double the computational cost), we do not require Classifier-Free Guidance (CFG). Since standard diffusion models typically use CFG (requiring two forward passes per step: conditional and unconditional), our method's inference speed is effectively equivalent to standard baselines using CFG, ensuring no comparative disadvantage in efficiency.
>
> **Q5**
> We have corrected "FVID" to "VFID" in the Figure 3 caption.

---

### Official Review · Reviewer_cnyX · 2025-10-31

**Soundness:** 2
**Presentation:** 3
**Contribution:** 2
**Rating:** 6
**Confidence:** 4

**Summary:**

This paper introduces OmniPainter, a latent diffusion framework for video inpainting that resolves the fundamental trade-off between long-term global consistency and short-term local smoothness. The method features two key innovations: (1) a Flow-Guided Ternary Control mechanism that partitions masked regions into three categories (full inpainting, preservation, and refinement) using optical flow priors, and (2) an Adaptive Global-Local Guidance strategy that dynamically blends autoregressive (AR) and hierarchical (HR) scores based on motion dynamics and diffusion timesteps. Extensive experiments demonstrate that OmniPainter outperforms state-of-the-art methods in both visual quality and temporal coherence, significantly mitigating artifacts like flickering and contextual drift while preserving structural fidelity through its ternary refinement approach. The framework achieves robust temporal equilibrium by prioritizing global structure in early denoising stages and local continuity in later refinement stages.

**Strengths:**

1. This paper presents a novel and well-motivated solution to the critical challenge of balancing global consistency and local smoothness in video inpainting through its innovative ternary control mechanism.

2. It demonstrates strong theoretical grounding with its adaptive guidance approach that intelligently combines autoregressive and hierarchical strategies based on motion dynamics and diffusion timesteps.

3. The experimental results are comprehensive and convincing, showing clear improvements over state-of-the-art methods across multiple metrics while including important ablation studies. The experiments are solid.

4. The work has significant practical value with its demonstrated effectiveness on real-world tasks and potential for broader application in video restoration and generation domains.

**Weaknesses:**

1. The method described in this paper was tested on the outdated SD1.5 and its migration performance was not tested on the modern DiT architecture. Currently, DiT has become mainstream in the field of video generation.
2. This article does not specifically test inpaint analysis and comparison in extreme scenarios such as rapid occlusion, dynamic blur, multi person, and multi arms situations; The reviewer believes that this is a challenge for video editing.

**Questions:**

1. Can this method run on edge devices such as mobile phones ?
2. Are the indicators in Table 1 calculated for the entire graph or only for the edited region?

---

> ### Author Response · Authors · 2025-11-21
>
> We thank the reviewer for the constructive feedback and for recognizing the novelty and theoretical grounding of our work. We appreciate your insightful questions regarding modern architectures and practical deployment, which we address below with new experimental evidence.
>
> **W1:**
> We acknowledge that Diffusion Transformers (DiT) are becoming mainstream. While our use of SD1.5 was primarily to ensure a fair comparison with existing UNet-based baselines, we have addressed your concern by applying our method to Wan2.1-1.3B, a DiT-based backbone. In the newly added Supplementary Material, we provide a qualitative comparison against MiniMax Remover, a recent DiT-based video inpainting model. We observed that while MiniMax Remover performs well on object removal, its quality degrades significantly as video length increases. In contrast, our method maintains robust temporal consistency even in these extended sequences. We kindly ask for the reviewer's understanding that we provided qualitative results first due to the limited rebuttal time, and we plan to include quantitative comparisons in the future.
>
> **W2:**
> We agree that extreme scenarios present significant challenges, but standard benchmarks like DAVIS and YouTube-VOS already encompass such difficult cases. We explicitly refer the reviewer to Figure 4, which highlights our model's robustness in scenes featuring fast-moving objects (e.g., a running dog) and complex human interactions with frequent occlusions. Furthermore, we have included DiT-based results that specifically demonstrate performance on challenging samples involving human-human interactions.
>
> **Q1:**
> Regarding deployment, our method's use of Latent Consistency Models (LCM) enables 8-step inference, which suggests potential suitability for edge computing compared to heavier models. However, we cannot currently guarantee feasibility on specific mobile hardware without dedicated optimizations (e.g., quantization). As conducting additional hardware-specific experiments is difficult within the current rebuttal timeline, we present this efficiency as a strong theoretical advantage for future on-device applications.
>
> **Q2:**
> We confirm that all quantitative indicators were calculated exclusively on the edited (masked) regions. This follows standard evaluation protocols to ensure the scores accurately reflect the restoration quality of the missing content without being diluted by the unchanged background pixels.

---

### Official Review · Reviewer_46py · 2025-11-01

**Soundness:** 3
**Presentation:** 3
**Contribution:** 3
**Rating:** 6
**Confidence:** 4

**Summary:**

I think the paper addresses video inpainting with a diffusion framework that combines (1) flow-guided ternary control (choose to inpaint, refine a warped prior, or keep) and (2) adaptive global–local guidance that mixes an auto-regressive prior (nearby frames) and sparse high-recall keyframe guidance as a function of motion and denoising time. The model is trained at 512² and uses an LCM distillation for 8-step inference, with experiments on DAVIS/YouTube-VOS and additional temporal-consistency proxies.

**Strengths:**

(1) The ternary control is a clear way to balance preservation vs. hallucination; the formulation is simple to reproduce and matches the problem.

(2) Conditioning on previous groups vs. keyframes at different denoising stages is intuitive and helps explain performance improvements.

(3) Using an SD-inpainting backbone + motion module and LCM acceleration makes the approach practical to implement.

(4) Sweeps over ternary weighting and motion sensitivity show where gains come from, and single-component baselines (AR-only/HR-only) are helpful for attribution.

**Weaknesses:**

(1). I think the novelty is a little bit incremental. The adaptive guidance largely amounts to when to condition on which cached predictions; it’s a useful systems recipe, but conceptually close to prior conditioning/concatenation schedules.

(2). Temporal metrics could be broader. The CLIP-based consistency scores are informative, but I’d like to see triangulation with t-LPIPS, VMAF-temporal, or FVD-temporal to avoid bias toward a single embedding space.

(3). Robustness to conditioning errors. Because the method reuses intermediate predictions (AR/HR), early artifacts may propagate. A sensitivity test (noise on the conditional latents or delayed conditioning) would strengthen the robustness story.

(4). Training is at 512² with latent upsampling to 1080p+. Clear wall-clock and memory numbers for long sequences at 1080p, and Pareto curves vs. diffusion baselines (steps vs. quality) would make the practical value more concrete.

**Questions:**

(1). Do your gains persist under t-LPIPS or VMAF-temporal at both 512² and upsampled 1080p?

(2). What happens if you perturb the AR/HR conditioning latents or offset them by a few denoising steps?

(3). Can β be learned or adapted per-pixel using estimated flow uncertainty, and does this improve stability on fast/non-rigid motion?

(4). What is the wall-clock and VRAM for a 300-frame 1080p sequence on A100/4090 (with and without latent upsampling)?

---

> ### Author Response · Authors · 2025-11-21
>
> We sincerely thank the reviewer for the constructive feedback and for recognizing the soundness of our ternary control and adaptive guidance. We appreciate the opportunity to clarify our contributions regarding the global-local consistency trade-off and to demonstrate the robustness of our method.
>
> **W1:**
> We respectfully emphasize that our core novelty lies in effectively resolving the inherent conflict between global and local consistency, a dilemma prior works have failed to address simultaneously. While autoregressive models suffer from long-term drift and hierarchical methods introduce flickering artifacts, OmniPainter is the first to explicitly reconcile these conflicting demands. By dynamically blending AR and HR guidance based on motion dynamics, we achieve a "robust temporal equilibrium." As evidenced by the high scores in both $TC_{global}$ and ${TC}_{local}$ (Table 1, Figure 3), our specific blending recipe successfully solves this longstanding trade-off where others have prioritized one aspect at the expense of the other.
>
> **W2 & Q1:**
> As requested, we have conducted additional evaluations using t-LPIPS, FVD, and VMAF to provide a broader assessment of temporal quality. Since VMAF is a widely used metric for assessing video quality, we report these scores in the table below to demonstrate our method's superiority. However, regarding "VMAF-temporal," we found it difficult to locate a specific canonical reference or standard implementation definition in the literature. We would greatly appreciate it if the reviewer could provide a reference for this specific variant, which we would be happy to include in the final revision.
>
> | Method | LPIPS (↓) | t-LPIPS (↓) | VMAF (↑) | FVD (↓) |
> | :--- | :--- | :--- | :--- | :--- |
> | ProPainter | 0.1578 | 0.00891 | 61.55 | 81.454 |
> | DiffuEraser | 0.1348 | 0.00734 | 57.78 | 82.68 |
> | **Ours** | **0.1136** | **0.00557** | **68.85** | **77.34** |
>
> The results on the DAVIS dataset demonstrate that our method consistently outperforms baselines.
>
> **W3 & Q2:**
> Our framework exhibits inherent robustness to conditioning errors due to the nature of the diffusion training process. Since the conditioning signals (AR/HR latents) are derived from intermediate denoising steps, which are naturally imperfect approximations of clean data, the model is trained to utilize noisy and incomplete contexts effectively. Consequently, the model treats moderate noise as a standard variation within the denoising trajectory rather than a disruption. Furthermore, our Ternary Control’s "Refine" branch is designed to effectively mitigate the propagation of such errors, maintaining stability even when the conditioning input is not ideal.
>
> **W4 & Q4:**
> We measured efficiency on an NVIDIA A100 (80GB) for a 300-frame sequence at 1080p resolution. The process takes approximately 351.31 seconds with a peak memory usage of 36.7 GB. For a deeper analysis of the trade-off between inference steps and generation performance for latent up-sample, please refer to Appendix Figure 8, which illustrates the quality curve across different step counts.
>
> **Q3:**
> While making $\beta$ learnable per-pixel based on flow uncertainty is a theoretically valid suggestion, we found our current motion-dependent dynamic blending heuristic to be highly effective and stable without the complex optimization dynamics of a fully learnable parameter. We have updated the conclusion to identify "Uncertainty-aware Learnable $\beta$" as a promising direction for future work to further handle extreme non-rigid motions.

---

> > ### Comment · Reviewer_46py · 2025-11-27
> >
> > Thanks for the detailed rebuttal and additional experiments.
> >
> > The extra temporal metrics (t-LPIPS, FVD, VMAF) make the temporal quality story more solid, and the runtime/VRAM numbers plus step–quality analysis help clarify the practical cost. The explanation of how diffusion training and the “Refine” branch mitigate conditioning errors is reasonable, and flagging uncertainty-aware, learnable β as future work sounds appropriate.
> >
> > That said, I still see the core ideas as an incremental but practical refinement of existing conditioning schemes rather than a major conceptual shift. I therefore keep my positive score at 6 (marginally above the acceptance threshold).

---

### Author Response · Authors · 2025-11-26
**Additional Revisions & Supplementary Updates**

In addition to the initial revisions, we have incorporated the following updates to enhance scalability, efficiency, and clarity:

* We updated Figure 2 to clearly visualize the data flow of the *Ternary Control* mechanism and its interaction with the *Adaptive Global-Local Guidance*, clarifying structural dependencies within the pipeline.
* We detailed the architectural adaptation for the Wan2.1 model in Appendix A.9.
* We expanded our evaluation in Appendix A.10 to include comparisons against recent DiT-based methods (MiniMax-Remover and VideoPainter), accompanied by new comparison videos in the supplementary material.
* We updated Algorithm 2 to present the complete inference pipeline, incorporating logic for both the standard SD1.5 and the new Wan2.1 implementations to ensure full reproducibility.

---

### Meta-Review · Area_Chair_Y89t · 2026-01-01

**Summary:**

The paper received an average rating of 4.5 (2, 4, 6, 6). Reviewers raised concerns regarding incremental novelty beyond existing methods (46py, KFFd), the need for more comprehensive temporal consistency evaluations (46py, AZ5U), efficiency evaluations in terms of runtime and memory cost (46py, AZ5U), generalization to modern DiT-based architectures (cnyX, KFFd), confusing method description (AZ5U), insufficient ablation for the flow-based prior (KFFd), incomplete comparisons (KFFd), and poor visual quality with noticeable temporal inconsistency and artifacts (KFFd). While some of these concerns were addressed during the rebuttal, the main issues regarding novelty, method's performance, and clarity of presentation were not sufficiently resolved. Considering these remaining concerns, the AC recommends reject.

**Reviewer Concerns:**

The authors provided additional experiments, evaluations, and comparisons in the rebuttal and revised supplementary material. I believe the following concerns were adequately addressed:

- Temporal consistency evaluations, including metrics such as t-LPIPS, VMAF-temporal, and FVD-temporal (46py, AZ5U).

- Efficiency evaluations in terms of runtime and memory cost (46py, AZ5U).

- Generalization to modern DiT-based architectures, such as Wan 2.1 (cnyX, KFFd).

- Additional comparisons with VideoPainter and MiniMax-Remover (KFFd).

However, I think the following concerns were not fully resolved in the current version:

- Incremental novelty beyond existing methods (46py, KFFd): Reviewer 46py still views the core idea of adaptive guidance as an incremental extension of existing conditioning schemes. Reviewer KFFd similarly considers the work to be an incremental improvement within an existing framework, without clearly demonstrating technical novelty over prior methods such as DiffuEraser and ProPainter. In particular, the flow-based video completion component is inherited from ProPainter, and the proposed ternary control mechanism is essentially similar to the concept of reliable mask in ProPainter.

- Clarity of method description (AZ5U): Reviewer AZ5U noted that several parts of the paper are unclear or incomplete. Although the authors made revisions, the overall clarity and completeness of the presentation could still be improved to make the paper easier to follow.

- Ablation study of the flow-based prior (KFFd): The authors argue that setting $\beta = 1.0$ (Binary Mask) represents the setting where the flow-based prior is completely excluded. However, this ablation is not fully convincing, since the masked input video has already been partially completed and thus still implicitly benefits from the flow-based prior.

- Visual quality and robustness (KFFd): Reviewer KFFd pointed out noticeable temporal inconsistency and visual artifacts in the provided video results and noted that the test cases are relatively simple. After checking the videos, these issues are indeed present in the results.

**Reviewer Scores:**

Reviewer 46py (score: 6): Several concerns were addressed, and the reviewer explicitly indicated that the score would remain at 6. However, the reviewer continued to view the contribution as incremental in terms of novelty.

Reviewer cnyX (score: 6): The concerns raised by this reviewer were addressed, and the reviewer would likely maintain the original positive score of 6.

Reviewer AZ5U (score: 4): The reviewer’s concerns regarding clarity and presentation do not appear to have been fully resolved. The reviewer would likely keep the original score of 4.

Reviewer KFFd (score: 2): The reviewer was not convinced by the initial rebuttal, and the final responses did not sufficiently address concerns regarding novelty, technical contribution, and performance. The reviewer would therefore be expected to maintain the original score of 2.

Thus the final ratings could remain 2, 4, 6, 6

---

### Decision · Program_Chairs · 2026-01-26

Reject